# Watermark Smoothing Attacks against Language Models

## Abstract

Statistical watermarking is a technique used to embed a hidden signal in the probability distribution of text generated by large language models (LLMs), enabling the attribution of the text to the originating model. We introduce the *smoothing attack* and show that existing statistical watermarking methods are not robust against minor modifications of text. In particular, with the help of a weaker language model, an adversary can smooth out the distribution perturbation caused by watermarks. The resulting generated text achieves comparable quality to the original (unwatermarked) model while bypassing the watermark detector. Our attack reveals a fundamental limitation of a wide range of watermarking techniques.

## 1 Introduction

Large language models (LLMs) have made remarkable progress, posing the challenge of determining whether a text is written by a human or generated by AI (more specifically, by a given LLM). One common solution to this challenge is watermarking the generated text (Aaronson, 2023; Christ et al., 2023; Huang et al., 2023; Li et al., 2024), where the model provider subtly modifies the probability distribution of the generated text (i.e., a sequence of tokens). For example, at each position of token generation, the likelihood of selecting from a subset of tokens (referred to as the "green list") is slightly boosted, where the assignment of this subset is kept secret from the human users. This subtle statistical modification, while remaining largely unnoticeable to human users, can be observed by the detector who knows the secret and attributed to the watermark (Kirchenbauer et al., 2023a;b; Zhao et al., 2023a; Kuditipudi et al., 2023). Watermarking faces two main technical challenges, maintaining text quality and preventing easy removal of the watermark. In this work, we focus on the second challenge and examine the resilience of statistical watermarking against attacks.

Previous attacks typically rely on a stronger/larger reference model to erase watermarks, e.g., using a strong GPT model to paraphrase the watermarked texts generated from Llama2-7b (Kirchenbauer et al., 2023b; Zhao et al., 2023a; Pan et al., 2024; Piet et al., 2023; Jovanović et al., 2024). However, the assumption that the attacker has access to a stronger model undermines the realism of the attack scenario, as such resources may not always be available in practice.

In this paper, we attack statistical watermarking of LLMs under more practical conditions. In our setting, the attacker's goal is not simply to remove the watermark at *any* cost, e.g., using a stronger model to paraphrase the watermarked text. Instead, we consider a more realistic attack scenario, where the adversary only has access to a weaker reference language model. The central question is: *given access to a weaker model, can the attacker remove the watermark while maintaining the quality of the text?*

We provide a positive answer in this paper, by presenting an attack algorithm, called the *watermark smoothing attack*. Our attack, which only queries the target watermarked model and a weaker reference model through model APIs, is able to remove the watermark from the generated text while maintaining its quality comparable to the original unwatermarked text. As suggested by its name, the key component of our attack is to smooth the shift of token distributions caused by watermarking. To do that, at each token position, we sample from a mixture of token distributions from the reference model and the watermarked model, where the coefficient of the mixture depends on the level of significance of the watermark. A higher level of significance means that the change caused by the watermark is more noticeable; hence, we should assign a larger coefficient to the reference model to smooth out this change, thereby removing the watermark. Conversely, a lower level of sig-

nificance means that the change caused by the watermark is not noticeable, in the first place; hence, we should assign a larger coefficient to the watermarked model for better text quality.

The major advantage of our *smoothing attack* over existing ones, is that ours is *watermark-agnostic* as it effectively smooths out the distribution shift caused by watermarking, without knowing the exact set(s) of tokens that are more likely to get sampled and the exact level of increase in their likelihoods. As a result, our method can be applied off-the-shelf to attack *any* statistical watermarking algorithm that relies on boosting the likelihood of sampling from specific tokens.

We conduct comprehensive experiments to validate the effectiveness of our attack against eight existing representative watermark strategies on Llama2-7b (Touvron et al., 2023) and OPT-1.3b (Zhang et al., 2022). Notably, under certain setups, our attack removes the watermark completely, meaning that $0\%$ of the generated text is detectable while maintaining high text quality. Meanwhile, under the same setup, the previous state-of-the-art attack (Piet et al., 2023) that uses the strong GPT-3.5-turbo (OpenAI, 2023a) to paraphrase the watermarked text fails to remove the watermarks in $48\%$ of the text. We emphasize that our attack achieves this significant improvement with access to only much weaker reference models such as TinyLlama-1.3b (Zhang et al., 2024b), and OPT-125m (Zhang et al., 2022), confirming its practicality and effectiveness.

## 2 PRELIMINARIES AND PROBLEM STATEMENT

**Text generation of language models.** We use $M$ to denote a language model (LM) and $\mathcal{V}$ to denote its associated vocabulary set (namely, the collection of all tokens). The LM generates a sequence of tokens step by step, constituting the output text. During the generation, at each token position $t$, $M$ computes the logit scores of tokens at position $t$, written as a vector $l_t \in \mathbb{R}^{|\mathcal{V}|}$ (where each dimension corresponds to the logit of a specific token); after that $M$ applies the softmax function (Bridle, 1989) to $l_t$, obtaining a probability distribution from which the $t$-th token is sampled. Two strategies are often employed at this sampling step, top-k sampling (Fan et al., 2018; Holtzman et al., 2018), which samples a token from the $k$ most probable candidates, and top-p/Nucleus sampling (Holtzman et al., 2019), which selects a token from the smallest set of tokens whose cumulative probability exceeds a threshold $p$.

**Statistical watermarking.** The overall idea of statistical watermarking for LMs (Kirchenbauer et al., 2023a;b; Zhao et al., 2023a) is to introduce a small shift to the probabilities for sampling a specific set of tokens (which is not revealed to the users), in such a way that a detector algorithm can identify this shift from a long text, whereas human users cannot. The canonical approach is as follows. At each token position $t$, a $\gamma$ fraction of the vocabulary set is first selected as the green list (denoted as $\mathcal{G}_t$); next, the logit values for tokens in $\mathcal{G}_t$ are increased by $\delta$ while the logit values for the rest of the vocabulary remains unchanged.

$$\tilde{l}_t^v = l_t^v + \delta \cdot \mathbb{1}_{v \in \mathcal{G}_t}. \tag{1}$$

The next token is then sampled according to the shifted logit vector $\tilde{l}_t$. In this way, a statistical watermark is embedded into the generated text, as the token distributions at all positions are now biased toward tokens in the green list(s). We denote this associated watermarked LM as $\tilde{M}$.

Provided with the assignment of $\mathcal{G}_t$, the detector algorithm looks for the evidence that tokens in $\mathcal{G}_t$ appear *disproportionally more frequent*. In particular, given the generated text of length $T$, the detector counts the actual occurrences of green tokens $n_{\text{green}}$ and computes the z-score as

$$z = \frac{(n_{\text{green}} - \gamma T)}{\sqrt{T\gamma(1-\gamma)}}, \tag{2}$$

and then predicts the given sequence as watermarked when $z$ exceeds some threshold $\lambda$ (namely, when $n_{\text{green}}$ exceeds $\gamma T$ by a large margin).

**Green token assignment.** The assignment of the green token set $\mathcal{G}_t$ is random at each token position $t$, determined by the LM provider. Due to this, it is very unlikely that a non-watermarked text will be misclassified as watermarked, particularly when $T$ is long enough. In addition, the assignment of $\mathcal{G}_t$ could also depend on previous $h$ tokens in the previously generated tokens (i.e., the prefix). For example, when $h = 0$, the assignment is context-independent and is referred to as the *Unigram* watermark (Zhao et al., 2023a); when $h = 1$, the assignment depends only on the previous token

and is referred to as the *KGW* watermark (Kirchenbauer et al., 2023a). Finally, we remark that $\mathcal{G}_t$ is not revealed to the LM users.

**Problem statement.** We present an attack to analyze the robustness of statistical watermarking. We assume the attacker has API access to a reference model $M_{\text{ref}}$, which is weaker compared to the original LM $M$; since otherwise, he would have less motivation to *attack* the watermarked model, which rules out the paraphrasing attacks that leverage a strong LM (e.g., ChatGPT) to paraphrase the watermarked text (Zhang et al., 2024a). Regarding the knowledge of the watermarked model $\tilde{M}$, we consider the realistic scenario that the adversary is unaware of the specific statistical watermarking strategies in use, e.g., the adversary does not know whether the assignment of $\mathcal{G}_t$ is context-dependent or not. The adversary can obtain the token sampled from the $\tilde{M}$ and $M_{\text{ref}}$, as well as the log probabilities of the most probable $K$ tokens. This level of information is commonly available, even when the model is closed-sourced, e.g., through OpenAI's API [1].

## 3 ATTACK FRAMEWORK

Recall Section 2 that the idea of statistical watermarking is to increase the sampling probability for a certain fraction of tokens (namely, the green tokens), by shifting their logit values. From a detector's point of view, the trace of the watermark is said to be *significant*, when the *actual occurrence* of green tokens in the generated text from the watermarked model are significantly higher than any other unwatermarked model (which is roughly $\gamma T$ for texts of length $T$).

Our attack exploits the above observation to bypass a detector. Consider the following extreme scenario(s). When generating the token for position $t$, the unwatermarked model already assigns high logit values for some green tokens (alternatively, some red tokens), possibly due to the inherent characteristics of the model or just by chance. In such scenarios, we would not expect that adding $\delta$ to the logit values of the green tokens that are too high (or too low) will cause a significant difference in the actual occurrence of a green token at this position. In other words, this $\delta$ shift is not likely to increase the z-score. Accordingly, as an attacker, we can just use the watermarked model for this position without modifying the output distribution. Conversely, in the not-so-extreme scenarios where the logit values for the green tokens are neither too high nor too low, the $\delta$ shift applied to the green tokens could have a significant influence on the z-score. In such cases, we smooth out this influence, by combining the token distributions of the watermarked model with those of a reference model to generate the next token.

Overall, our attack runs as follows. When generating the token at position $t$, **(i)** we first estimate the *significance level* of the watermark, which is defined as the relative increase in the probability of generating green tokens from the watermarked model compared to the unwatermarked model; **(ii)** if the significance level is high, then we combine the token distributions of the watermarked model and the reference model to generate the $t$-th token; otherwise, we generate the token from the watermarked model directly. Note that our attack is not designed for any statistical watermarking scheme in specific. Instead, our attack is universally applicable to all statistical watermarking schemes, as it directly aims at smoothing out the change in the actual occurrence of green tokens caused by the watermark. We describe our attack in more detail next.

### 3.1 SIGNIFICANCE LEVEL OF WATERMARKING

**Definition.** When sampling the token at position $t$, we are interested in how much more likely the watermarked model is to sample green tokens compared to the unwatermarked model. The natural definition is the probability difference across all green tokens: $\sum_{v \in \mathcal{G}_t} \tilde{p}_t^v - \sum_{v \in \mathcal{G}_t} p_t^v$, where $\tilde{p}_t^v$ and $p_t^v$ represent the probability of sampling token $v$ at position $t$ from the watermarked model $\tilde{M}$ and unwatermarked model $M$, respectively. However, summing over all green tokens may not accurately reflect the likelihood of sampling a green token, particularly due to the large number of tokens in $\mathcal{G}_t$. In particular, although the size of $\mathcal{G}_t$ could be as large as $16,000$ for Llama2 models when $\gamma = 0.5$, in practice, the next token is often chosen using top-k or top-p sampling (recall Section 2), where only the most probable tokens could be sampled in the first place (instead of all $16,000$ tokens in $\mathcal{G}_t$). Motivated by this, we focus on the $K$ most probable tokens. We denote the

---

[1]https://platform.openai.com/docs/advanced-usage/token-log-probabilities

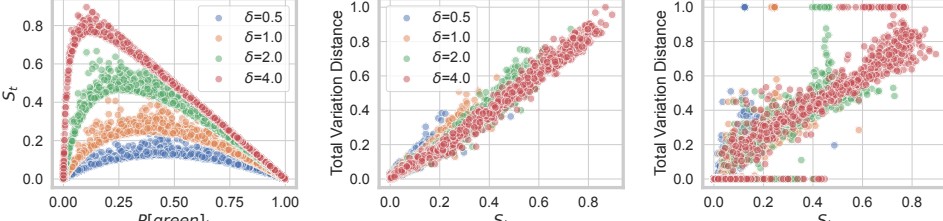

Figure 1: Correlations between $S_t$, $P_t[\text{green}]$, and the total variation distance (TVD) between the token distributions at position $t$ of the unwatermarked and watermarked models (with $\gamma = 0.5$), computed on OPT-1.3b. **Left** subfigure illustrates the correlation between the $S_t$ and $P_t[\text{green}]$–as $P_t[\text{green}]$ increases from 0 to 1, $S_t$ first increases and then decreases. **Middle** subfigure illustrates the positive correlation between $S_t$ and the TVD measured on the empirical token distributions of the watermarked and unwatermarked models, when top-k sampling ($k = 10$) is applied. **Right** subfigure leads to the same conclusion as the middle one when top-p sampling ($p = 0.8$) is applied.

sets of tokens that rank as the $K$ most probable ones according to $\tilde{p}_t$ and $p_t$ as $\mathcal{V}_{\tilde{p}_t}(K)$ and $\mathcal{V}_{p_t}(K)$, respectively. Accordingly, we define the probabilities of sampling green tokens at position $t$ from the watermarked model and unwatermarked model as

$$\tilde{P}_t[\text{green}] := \frac{\sum_{v \in \mathcal{G}_t \cap \mathcal{V}_{\tilde{p}_t}(K)} \tilde{p}_t^v}{\sum_{v \in \mathcal{V}_{\tilde{p}_t}(K)} \tilde{p}_t^v} \text{ and } P_t[\text{green}] := \frac{\sum_{v \in \mathcal{G}_t \cap \mathcal{V}_{p_t}(K)} p_t^v}{\sum_{v \in \mathcal{V}_{p_t}(K)} p_t^v}, \quad (3)$$

respectively. We proceed to define the significance level $S_t$ as

$$S_t := \tilde{P}_t[\text{green}] - P_t[\text{green}]. \quad (4)$$

In the rest, we consider the case where $K = 20$ for measuring $S_t$. We next explain the importance of $S_t$ in building our attack through illustrative observations.

**How $P_t[\text{green}]$ influences $S_t$.** Our first observation is that the probability of sampling green tokens from the unwatermarked model (i.e., $P_t[\text{green}]$) influences the value of the significance level (i.e., $S_t$). The left of Figure 1 illustrates the correlation between them. Each sample is obtained by querying the watermarked and unwatermarked models using the same prefix, from which we can compute $P_t[\text{green}]$, $\tilde{P}_t[\text{green}]$, and $S_t$. Overall, when $P_t[\text{green}]$ is either too large (close to 1) or too small (close to 0), $S_t$ is not significant; conversely, when $P_t[\text{green}]$ is of some moderate value (e.g., within the range of $[0.2, 0.8]$), $S_t$ becomes relatively more significant, resulting in a bell-shaped curve. Our observation holds under different choices of watermark shift $\delta$.

**How $S_t$ influences the trace of watermarking.** In the middle and right subfigures of Figure 1, we further show that when $S_t$ is large, the probability distributions for generating the token at position $t$ using the watermarked model and unwatermarked model are more different. We empirically measure this difference using the total variation distance (TVD) between the actual frequencies of tokens generated from the watermarked model and the unwatermarked model (each frequency histogram consists of 1000 tokens generated from 1000 independent runs). We see that when $S_t$ is large, there is less agreement between the watermarked and unwatermarked models (i.e., larger TVD between their token distribution); and vice versa. Therefore, to remove the watermarking trace, the attacker should focus on smoothing the TVD at token positions with relatively large $S_t$. Conversely, to maintain text quality, they should use the watermarked model directly at token positions with small $S_t$, since the TVD between the watermarked and unwatermarked models is low, indicating little to no detectable watermarking trace.

**How to compute $S_t$.** Our goal is to estimate $S_t$ in the absence of the knowledge of the assignment of green tokens (namely, $\mathcal{G}_t$), which the attacker does not have access to. In what follows, we first present an indicator for $S_t$ and then show how to use the indicator to estimate $S_t$.

We first compute the difference in sampling probability between the most likely token and the $K$-th most likely token in the set $\mathcal{V}_{p_t}(K)$ as follows

$$\Delta_t^K = \max_{v \in \mathcal{V}_{\tilde{p}_t}(K)} \tilde{p}_t^v - \min_{u \in \mathcal{V}_{\tilde{p}_t}(K)} \tilde{p}_t^u \quad (5)$$

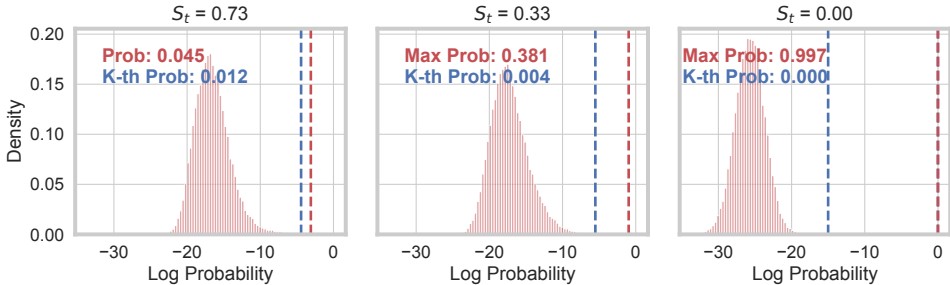

Figure 2: Distribution of the log of the probabilities of tokens from the watermarked OPT-1.3b (with $\gamma = 0.5, \delta = 2.0$). The vertical line represents the probability of the most probable token (0.997 means the probability is close to 1) and the 20-th most probable token. We can see that larger values of $S_t$ correspond to smaller maximum probabilities and smaller distances between the probabilities of the most probable token and the 20-th.

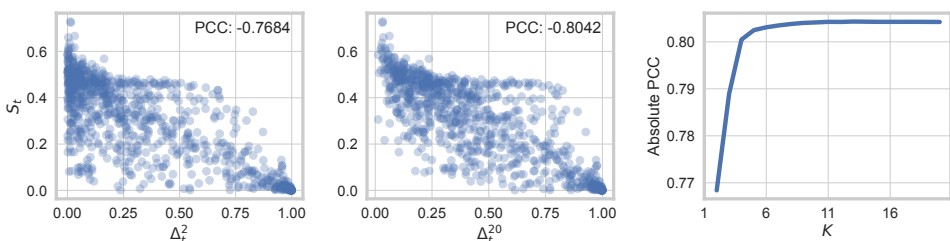

Figure 3: **Left and Middle** subfigures show the correlation between significance level $S_t$ and $\Delta_t^K$, computed on the watermarked OPT-1.3b (with $\gamma = 0.5$, $\delta = 2.0$) for $K = 2$ and $K = 20$, respectively. PCC stands for Pearson correlation coefficient, where larger values indicate stronger correlations. **Right** subfigure shows the absolute value of PCC with different choices of $K$.

We explain the intuition next. When $P_t[\text{green}]$ is extremely large or small, it is likely that there is a single token that "stands out" as the dominant token, which has a very high probability of getting sampled compared to others. Figure 2 illustrates the distributions of log probabilities of the watermarked model with different values of $S_t$, along with the maximum probability among all the tokens (i.e., $\max_{v \in \mathcal{V}_{\tilde{p}_t}(K)}$). We can see that, in general, smaller $S_t$ corresponds to larger maximum probabilities for the watermarked model. Intuitively, this is because when a single token has an extremely high probability of getting sampled, then adding delta to the logit values of other less likely tokens does not increase their chances of getting sampled significantly (namely, small $S_t$). In addition, adding $\delta$ to the logit value of that token does not change much of its probability of getting sampled either, as it is already very large (e.g., a value close to 1). In equation 5, we have subtracted the probability of the $K$-th most probable token from the most probable one. This can be seen as some sort of normalization, which cancels out the instability of the max probability (in our early experiments without this subtraction, the attack performance is not satisfactory). Figure 3 illustrates the strong correlation between $S_t$ and $\Delta_t^K$, with different values of $K$, confirming that $\Delta_t^K$ is a indicator for $S_t$ ($\Delta_t^K$ decreases as $S_t$ increases).

**Confidence score.** We now convert the $\Delta_t^K$ into a confidence score $c_t$ within the range $[0, 1]$. We start by querying the watermarked model on a set of prefixes to determine the upper bound and lower bound of $\Delta_t^K$, denoted as $U$ and $L$, respectively. (Alternatively, the attacker can simply use 0 and 1, to avoid making extra queries to the model.) Next, we divide the range between them into 100 bins of the same width. In particular, the $m$-th bin ($m = 1, \ldots, 100$) contains values in the range of $\left[ L + \frac{U-L}{100}(m-1), L + \frac{U-L}{100}m \right)$. At each token position $t$, the attacker calculates the $\Delta_t^K$ by querying the watermarked model and determines the corresponding bin index $i$ to put $\Delta_t^K$ in. The confidence score is the bin's relative position among the 100 bins, computed as $c_t = i/100$. A large value of $c_t$ is the result of a large $\Delta_t^K$, corresponding to a small significance level $S_t$. In this case, there is a high agreement between the watermarked model and the unwatermarked model

(recall Figure 1); hence, the attacker should have high *confidence* in using the watermarked model to sample the $t$-th token. Conversely, a small value of $c_t$ is the result of a small $\Delta_t^K$, corresponding to a large $S_t$. In this case, the attacker should have *less confidence* in using the watermarked model; instead, he should smooth out the change in probability distribution caused by the watermark shift.

### 3.2 Smoothing the Token Distribution

We now introduce our smoothing method. Specifically, given the confidence score $c_t$ (which is *inversely related* to the significance level $S_t$), the attacker samples the token from the watermarked model with probability $c_t$. Otherwise, the attacker samples a token from the reference model $\mathcal{M}_{\text{ref}}$. Accordingly, we can view probability distribution for sampling the next token as a mixture of $\tilde{p}_t$ (corresponding to the watermarked model) and $p_t^{\text{ref}}$ (corresponding to the reference model), written as

$$p_t^{\text{att}} = \begin{cases} \tilde{p}_t & \text{with probability } c_t \\ p_t^{\text{ref}} & \text{with probability } (1 - c_t) \end{cases}$$

The probability of sampling the green tokens using our method at position $t$ is then computed as

$$P_t^{\text{att}}[\text{green}] = c_t \cdot \tilde{P}_t[\text{green}] + (1 - c_t) \cdot P_t^{\text{ref}}[\text{green}], \tag{6}$$

where $\tilde{P}_t[\text{green}]$ is defined in Equation 3 and $P_t^{\text{ref}}[\text{green}] = \sum_{v \in \mathcal{G}_t \cap \mathcal{V}_{p_t^{\text{ref}}}(K)} p_t^{\text{ref},v} / \sum_{v \in \mathcal{V}_{p_t^{\text{ref}}}(K)} p_t^{\text{ref},v}$ is the expected probability of sampling green tokens based on the top $K$ tokens from the reference model. The difference between $P_t^{\text{att}}[\text{green}]$ and $P_t[\text{green}]$ is computed as

$$c_t \cdot \underbrace{\left( \tilde{P}_t[\text{green}] - P_t[\text{green}] \right)}_{S_t} + (1 - c_t) \cdot \left( P_t^{\text{ref}}[\text{green}] - P_t[\text{green}] \right). \tag{7}$$

Since the reference model is free of watermarks, its expected occurrence of green tokens should be similar to that of the unwatermarked model, making the second term small. Moving to the first term, we note that it attains a relatively large value when both $c_t$ and $S_t$ are large. This, however, is not possible due to our design–$c_t$ is *inversely related* $S_t$. In conclusion, the probability of sampling a green token using our $p_t^{\text{att}}$ is similar to the unwatermarked model; hence, our attack is able to bypass the watermark detector. We outline the complete algorithm as in Algorithm 1 in Appendix A.1.

To further ensure that we always bypass the watermark detector, we can set the probability of using the watermarked model to 0 (instead of $c_t$) 0 whenever $c_t$ is not *significant* enough, i.e., $c_t \leq \tau$. This ensures that the fraction of green tokens generated is closer to any model without watermarks. In Section 4, we demonstrate the impact of $\tau$ on both text quality and the effectiveness of the attack. We conclude this section with some remarks on our design.

**Agnostic smoothing.** Our attack is *agnostic* to how the statistical watermark is embedded, e.g., Unigram watermark (Zhao et al., 2023b) or KGW watermark (Kirchenbauer et al., 2023a). Regardless of the strategy for choosing the green token set $\mathcal{G}_t$ and internal details of the watermarked model, we apply a universal smoothing over the watermarked model's token distribution by combining it with that of a reference model. Here the level of smoothness depends on the significance level of the watermark, i.e., how much the watermark model agrees with the unwatermarked model, which, again, can be estimated accurately without any knowledge of the model or watermarking strategy.

**Practicality.** Our attack is practical, as it leverages the reference model and watermarked model as black boxes. Besides, the reference model is much weaker than the target watermarked model. Our practical attack provides more insight into the robustness analysis of statistical watermarking–not only is it vulnerable under the paraphrasing attack (Piet et al., 2023) that leverages a strong model, but also not resilient against our smoothing attack where the attacker only gets help from a much weaker model that contains an order of magnitude fewer parameters (as we will see next in the experiment). Our findings call for stronger watermark strategies for LLMs.

## 4 Experiments

In this section, we evaluate our attack from two axes–how effective our attack is in terms of removing the watermarks and preserving the quality of the generated text. All experiments are conducted on two NVIDIA TITAN RTX GPUs with 24GB memory for each.

## 4.1 SETUP

**Models and Datasets.** We consider two commonly used models as the target model in the watermarking literature, Llama2-7b (Touvron et al., 2023) and OPT-1.3b (Zhang et al., 2022). When attacking models Llama2-7b and OPT-1.3b, we use TinyLlama-1.3b (Zhang et al., 2024b) and OPT-125m (Zhang et al., 2022) as the reference models, respectively. Following prior work (Kirchenbauer et al., 2023a; Pan et al., 2024), we use the C4 dataset (Raffel et al., 2020). Specifically, the first 30 tokens of texts serve as prompts, and the task is to generate the subsequent 200 tokens. The original C4 texts act as human-written examples, referred to as the *human-written* baseline.

**Watermark Algorithms.** We evaluate against eight representative watermarking algorithms, including KGW (Kirchenbauer et al., 2023a), Unigram (Zhao et al., 2023a), SWEET (Lee et al., 2023), UPV (Liu et al., 2023), EWD (Lu et al., 2024), SIR (Liu et al., 2024), X-SIR (He et al., 2024), and DIP (Wu et al.). We adopt the implementation in MarkLLM toolkit (Pan et al., 2024) for the above watermarking algorithms. The detailed description of these algorithms is in Appendix A.2. Results for KGW and Unigram are presented in the main paper; results for the remaining algorithms can be found in Appendix A.3. For the two algorithms highlighted, we follow standard configurations by setting the watermark shift $\delta$ to 2, the fraction of green tokens $\gamma$ to $0.5$, and the z-score threshold for watermark prediction to $4$.

**Attacks.** Attack baselines include **Word-D**, which randomly deletes a word at a specified ratio and **Word-S**, which randomly substitutes a word with its synonyms using WordNet (Miller, 1995). Following (Pan et al., 2024), we set the ratio to be $0.3$ for Word-D and to be $0.5$ for Word-S. We also include the strong baseline **P-GPT3.5** (Piet et al., 2023) that paraphrases the given text based on the GPT-3.5-turbo using the prompt: "Please rewrite the following text:". It is important to note that this attack assumes a significantly stronger attacker than ours (**Smoothing**). We also include the reference model (**Reference**) and the unwatermarked model (**Unwatermarkd**) in our comparison. For our smoothing attack, we set the threshold $\tau$ to $0.5$ by default, unless specified otherwise. Namely, whenever the confidence level is smaller than $0.5$, we always sample from the reference model rather than the mixture of the reference and unwatermarked model.

**Metrics.** To measure the effectiveness of watermark removal, we compute the z-score (defined in equation 2) on the generated text. Lower values indicate fewer traces of watermarks in the generated text and greater success for the attacker. We also report the positive prediction rate (PPR) when using the default threshold on the z-score, which denotes the proportion of generated text identified as watermarked. There are two types of texts, positive samples (those generated from the watermarked model with/without attacks) and negative samples (those written by humans or generated by unwatermarked text). PPR reflects the True Positive Rate when computing on positive samples and the False Positive Rate when computing on negative samples.

To measure the text quality, we follow prior work (Kirchenbauer et al., 2023a; Pan et al., 2024) and compute the perplexity using an oracle model. We use OPT-2.7b for texts generated by OPT models, and Llama2-13b for those from Llama models. We also measure the negative log-likelihood (loss) of the unwatermarked model on the generated text to assess how likely the unwatermarked model would have produced it. Smaller perplexities and losses indicate better text quality. We also utilize GPT-4 (OpenAI, 2023b) as an evaluator of accuracy, consistency, and style, scoring on a scale of 1 to 10, similar to the approach in (Jovanović et al., 2024), with higher scores indicating better performance. The prompt template used for evaluation is provided in Appendix A.4.

## 4.2 EFFECTIVENESS OF SMOOTHING ATTACK

Table 1 presents the overall result. Compared to Word-D and Word-S which do not utilize any additional model, our smoothing attack achieves a lower positive prediction rate, indicating that more generated texts successfully bypass detection. Additionally, our attack maintains high text quality, achieving lower perplexity and loss. Compared to the text generated by the reference model alone, the text generated by our attack also achieves a higher quality, which justifies the adversary's motivation to remove the watermark from the target model. Otherwise, if the reference model can achieve a high text quality, the adversary could simply use the reference model which is watermark-free instead of launching an attack.

Notably, in some cases, our attack can be far more effective at removing the watermark compared to the paraphrasing attack that relies on a much stronger reference model. Under the Unigram

Table 1: Effectiveness of smoothing attacks on OPT-1.3b and Llama2-7b models using KGW and Unigram watermark algorithms. The values are aggregated over 300 responses. A lower z-score and lower positive prediction rate (PPR) indicate a stronger attack. At the same time, the adversary seeks to maintain high text quality, aiming for lower perplexity and loss values. The lowest values across all attacks on the watermarked models are highlighted in bold. For the GPT-4 score, higher values indicate better quality.

| Model | Algorithm | Setting | Effectiveness | | Text Quality | | |
|---|---|---|---|---|---|---|---|
| | | | Z-Score ↓ | PPR ↓ | Perplexity ↓ | Loss ↓ | GPT-4 ↑ |
| OPT-1.3b | KGW | Human-written | 0.12 | 0.00 | 14.59 | 2.68 | 8.83 |
| | | Reference | 0.21 | 0.00 | 19.75 | 2.98 | 7.16 |
| | | Unwatermarked | 0.07 | 0.00 | 12.30 | 2.51 | 8.66 |
| | | Watermarked | 8.03 | 1.00 | 16.52 | 2.80 | 8.33 |
| | | Watermarked (Word-D) | 5.13 | 0.85 | 87.20 | 4.47 | 2.0 |
| | | Watermarked (Word-S) | 3.12 | 0.17 | 175.15 | 5.17 | 2.66 |
| | | Watermarked (P-GPT3.5) | 2.54 | 0.19 | **15.27** | **2.73** | 8.66 |
| | | Watermarked (Smoothing) | **1.49** | **0.00** | 17.91 | 2.89 | 7.33 |
| | Unigram | Human-written | -0.22 | 0.00 | 14.59 | 2.68 | 8.83 |
| | | Reference | -0.07 | 0.00 | 19.51 | 2.97 | 7.16 |
| | | Unwatermarked | -0.05 | 0.00 | 12.45 | 2.52 | 8.66 |
| | | Watermarked | 8.56 | 0.99 | 16.80 | 2.82 | 7.66 |
| | | Watermarked (Word-D) | 7.04 | 0.98 | 91.98 | 4.52 | 2.0 |
| | | Watermarked (Word-S) | 5.42 | 0.88 | 190.26 | 5.25 | 2.66 |
| | | Watermarked (P-GPT3.5) | 3.70 | 0.48 | **14.62** | **2.68** | 8.33 |
| | | Watermarked (Smoothing) | **1.52** | **0.00** | 18.26 | 2.90 | 7.33 |
| Llama2-7b | KGW | Human-written | -0.78 | 0.00 | 7.37 | 2.00 | 8.83 |
| | | Reference | 0.11 | 0.00 | 17.10 | 2.72 | 3.3 |
| | | Unwatermarked | -0.74 | 0.00 | 4.08 | 1.41 | 8.66 |
| | | Watermarked | 6.47 | 0.90 | 5.11 | 1.63 | 8.28 |
| | | Watermarked (Word-D) | 3.88 | 0.45 | 25.69 | 3.25 | 2.48 |
| | | Watermarked (Word-S) | 2.78 | 0.15 | 31.16 | 3.44 | 3.81 |
| | | Watermarked (P-GPT3.5) | 2.22 | 0.15 | 5.46 | 1.70 | 8.83 |
| | | Watermarked (Smoothing) | 2.00 | **0.10** | **3.40** | **1.22** | 5.25 |
| | Unigram | Human-written | -0.94 | 0.00 | 7.37 | 2.00 | 8.83 |
| | | Reference | -0.21 | 0.05 | 16.46 | 2.65 | 3.33 |
| | | Unwatermarked | -2.30 | 0.00 | 4.12 | 1.42 | 8.66 |
| | | Watermarked | 6.85 | 0.95 | 4.90 | 1.59 | 8.33 |
| | | Watermarked (Word-D) | 5.31 | 0.65 | 20.73 | 3.03 | 2.17 |
| | | Watermarked (Word-S) | 3.34 | 0.40 | 31.93 | 3.46 | 4.0 |
| | | Watermarked (P-GPT3.5) | 2.00 | **0.10** | 7.17 | 1.88 | 9.0 |
| | | Watermarked (Smoothing) | **0.08** | 0.15 | **3.36** | **1.21** | 4.5 |

watermark for the OPT-1.3b model, the paraphrasing attack using the gpt-3.5-turbo model results in a positive prediction rate of $0.48$, meaning nearly half of the paraphrased text is still detected as watermarked. In contrast, our smoothing attack achieves a $0.0$ positive prediction rate. Our results demonstrate that the vulnerability of the watermark is more severe than previously assumed, as an adversary can successfully remove the watermark using a much weaker model while still preserving the quality of the text. The results for six other watermark algorithms are similar, and are presented in Table 2 of Appendix A.3.

Figure 4 provides a more detailed illustration of the perplexity and z-score for the text generated under different attacks, with the unwatermarked model (Un-W) and watermarked model (W) as references. Each point in this Figure represents a sample text. As we can see, our smoothing attack significantly lowers the z-score compared to the watermarked model, while preserving text quality. Specifically, the text generated by the smoothing attack closely mirrors the unwatermarked model in terms of the distributions for both quality and z-score. On the other hand, other attacks incur a significant drop in the text quality, resulting in extremely high perplexity.

**Effect of watermark shift $\delta$.** Figure 5 shows the performance of our attack in terms of z-score and perplexity, compared with the watermarked model under different choices of watermark shift $\delta$. From the right subfigure, we notice that when the watermark shift increases, the z-score increases significantly for the watermarked text, making the detection easier. On the other hand, the

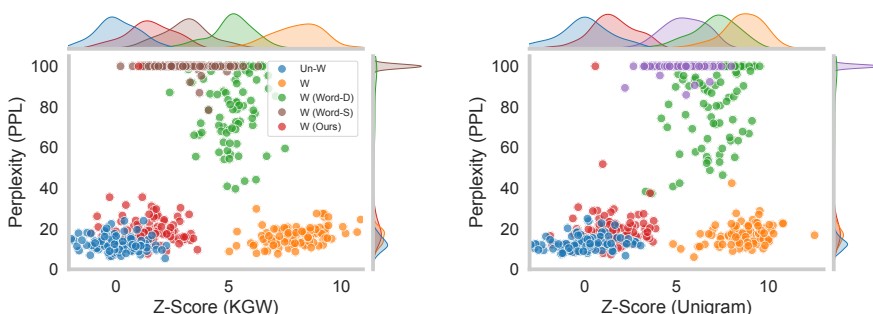

Figure 4: Effectiveness of smoothing attacks on the OPT-1.3b model with KGW and Unigram watermark algorithms. We compute the z-score and perplexity (with lower values indicating better quality) for 100 texts generated by the unwatermarked model (Un-W), watermarked model (W), and watermarked models under different attacks. For clarity, we truncate perplexity at an upper bound of 100. The texts generated by the smoothing attack remain closer to the unwatermarked texts in both perplexity and z-score compared to other attacks. Results for additional watermark methods are provided in Figure 7 in Appendix A.3.

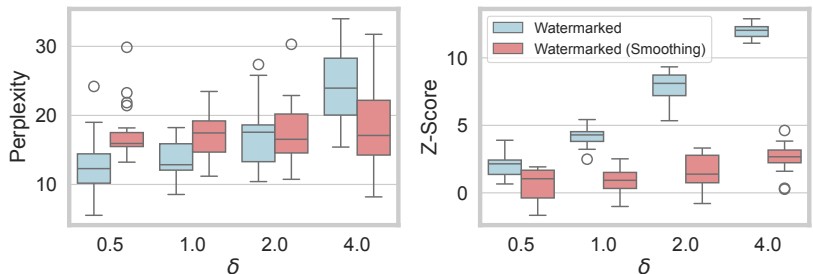

Figure 5: Effectiveness of the smoothing attack on OPT-1.3b with KGW watermarks under different watermark shift values, $\delta$. We present boxplots of the z-score and perplexity for responses from both the watermarked model and the smoothing attack model. Notably, the text quality from the attack model can surpass that of the watermarked model when the watermark shift is large.

z-score for the text generated by our attack model remains low, confirming the effectiveness of our attack, regardless of the assignment of $\delta$. From the left subfigure, we notice that the benefit of using smoothing in preserving the text quality increases as $\delta$ increases. That is, adding the watermark introduces a significant distortion to the output which hurts the text quality while our smoothing attack reduces this distortion, preserving the text quality. Notably, the text quality from the attack model can surpass that of the watermarked model when $\delta$ is large.

**Effect of $\tau$.** Figure 6 illustrates the performance of the smoothing attack in terms of z-score and perplexity across different values of $\tau$. Recall that the adversary only samples from the watermarked model when $c_t$ is sufficiently large, i.e., setting $c_t = 0$ when $c_t \leq \tau$. Therefore, with a higher $\tau$, the adversary is less likely to sample from the watermarked model. The results show that as $\tau$ increases, the z-score decreases, thereby lowering the likelihood of detection. Notably, increasing $\tau$ has a less noticeable impact on text quality, as the perplexity remains relatively stable.

## 5 RELATED WORKS

Early watermark solutions embed watermarks by altering text through synonyms (Topkara et al., 2005; 2006b), changing sentence structure (Topkara et al., 2006a), or injecting invisible tokens (Rizzo et al., 2019). More recent methods include context-aware lexical substitution (Yang et al., 2022) and mask-infilling models (Ueoka et al., 2021). Statistical watermarks that are based on "green-red" list increase the logits of tokens from the green list during token generation, thus altering the output token distributions (Zhao et al., 2023a; Kirchenbauer et al., 2023a; Lee et al., 2023; Liu et al., 2023; Lu et al., 2024; Liu et al., 2024; He et al., 2024; Wu et al.). On the contrary,

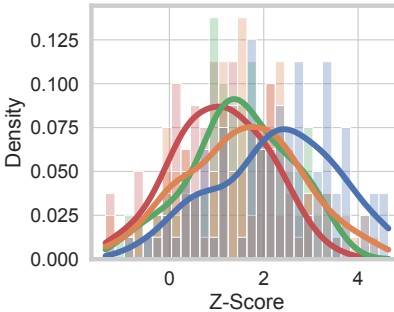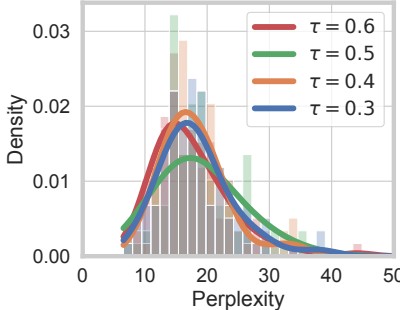

Figure 6: Effectiveness of smoothing attacks on the OPT-1.3b model with KGW watermarks with different $\tau$. We present histograms of the z-score and perplexity for responses generated by the smoothing attack. A larger $\tau$ leads to the exclusion of more tokens from the watermarked model, resulting in a decrease in the z-score, while the change in perplexity is less obvious.

distortion-free watermarks do not alter the output token distributions (Christ et al., 2023; Aaronson, 2023; Kuditipudi et al., 2023; Dathathri et al., 2024; Hu et al.) and hence, are inherently more vulnerable under our threat model, since the adversary can directly sample the next token from the watermark-free top-$K$ token probabilities.

We focus on evaluating the robustness of watermarking schemes via attacks. One canonical attack is to disrupt the structure of embedded watermarks, making them undetectable by injecting certain characters, homoglyphs, or emojis into the target text (Gabrilovich & Gontmakher, 2002; Helfrich & Neff, 2012; Pajola & Conti, 2021; Boucher et al., 2022; Goodside, 2023).

Another approach involves utilizing a separate large language model (LLM) to paraphrase the target text (Sadasivan et al., 2023; Krishna et al., 2023; Piet et al., 2023), where the paraphrasing LLM is *significantly larger* than the reference model targeted by our attack. Building on this, Jovanović et al. (2024) and Zhang et al. (2024a) introduce advanced methods that incorporate additional models to complement the paraphrasing LLM. Specifically, Jovanović et al. (2024) employs an auxiliary reference model to analyze how watermarks are embedded. Such analysis requires prior knowledge of the watermarking algorithm (e.g., the choice of $h$) as well as access to a large set of texts generated from the watermarked model. In contrast, our attack does not have such requirements. Zhang et al. (2024a) enhances paraphrasing attack by integrating it with a reward model (termed as quality oracle model), which evaluates candidates generated by the paraphrasing LLM (termed as perturbation oracle model). In particular, multiple candidates are generated for each target text and the reward model selects the best one. In comparison, our attack generates only a single candidate. Despite these differences, our attack—using only TinyLLM with 1.3 billion parameters—outperforms existing paraphrasing attacks that rely on much larger models, such as GPT-3.5 with 175 billion parameters, as demonstrated in Table 1. Thus, our attack proves to be not only practical and cost-efficient but also highly effective, illustrating that even a relatively low-resource adversary can bypass watermarking. These findings underscore the critical need for the development of more resilient watermarking strategies. Further details and comparisons are provided in Appendix A.5.

## 6 CONCLUSION

In this work, we present the smoothing attack to remove statistical watermarks that are embedded into LLMs. The core idea is to smooth out the increased probability of sampling green tokens due to the embedded watermark. Our method is agnostic to the specific watermarking algorithm used and relies on more realistic assumptions on the attacker. Through comprehensive evaluations, we demonstrated that our attack can successfully remove watermarks from a wide spectrum of watermarking algorithms while preserving the quality of the generated text. Our findings highlight vulnerabilities in existing statistical watermarking techniques.

Our attack removes watermarks at the cost of a quality drop in some cases, leaving room for future improvement. Besides, designing more advanced watermarking techniques for LLMs that are robust against our smoothing attack is also worth looking into.

**Ethics Statement**    We, the authors of this paper, have reviewed and adhered to the ICLR Code of Ethics throughout the development and submission of our work. In preparing this paper, we have taken the following ethical considerations into account:

- Human Subjects Involvement: Our research does not involve human subjects, and therefore, no IRB (Institutional Review Board) approval was required.

- Data Set Usage and Release: The data sets used in this work are publicly available and do not contain personally identifiable information.

- Harmful Insights, Methodologies, and Applications: While our research does not propose methods that could be directly misused (e.g., for privacy violations, unethical surveillance, or discrimination), we acknowledge that our findings highlight vulnerabilities in watermarking algorithms under weaker adversarial assumptions. If these watermark algorithms are used to prevent misuse of generative AI, our approach could potentially be exploited to bypass detection. However, we note that this vulnerability is already recognized in the existing literature.

- Bias and Fairness Considerations: Our method and evaluation do not involve demographic data, and therefore do not directly raise concerns related to bias or fairness.

- Conflict of Interest and Sponsorship: There are no conflicts of interest associated with the sponsorship or funding of this work. Any affiliations or financial support for this research have been transparently disclosed.

- Privacy and Security: This work does not involve the collection or handling of private or sensitive data.

- Legal Compliance and Research Integrity: All research conducted in this paper complies with relevant legal and ethical standards.

We are committed to ensuring that our research upholds the highest standards of ethical responsibility, and we welcome discussions about the ethical implications of our work.

**Reproducibility Statement**    Our code is built on MarkLLM Pan et al. (2024) and is distributed under the Apache License 2.0. Modifications to the original source code are clearly documented in the readme.md file. We have provided scripts to reproduce the results presented in this paper, along with detailed instructions for setting up the environment. A comprehensive description of the dataset and preprocessing steps is also included. Additionally, all necessary dependencies and configuration files are supplied to ensure accurate reproducibility of the experiments.

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

# A    SUPPLEMENTAL MATERIAL

---

**Algorithm 1** Watermark Smoothing Attack

---

1: **Input:** Sampled token from the watermarked model $v_t$, log of probability for the top-K tokens Top-K$(p_t)$, sampled token from reference model $v_t^{ref}$.
2: **Output:** Token at $t$.
3: Compute $\Delta_t^K$ based on Equation 5.
4: $c_t = \text{BinIndex}(\Delta_t^K)/100$.
5: Sample a bit with probability $c_t$ as 1 and $(1 - c_t)$ probability as 0.
6: **if** Bit is 1 **then**
7:    Return $v_t$
8: **else**
9:    Return $v_t^{ref}$
10: **end if**

---

## A.1    ALGORITHM

For completeness, we present the algorithm in Algorithm 1. The algorithm first computes $\Delta_t^K$, the difference between the most probable token and the K-th most probable token from the watermarked model. This difference is then converted into a confidence score $c_t$ ranging from 0 to 1. The adversary samples a bit randomly, with probability $c_t$ for 1 and $(1 - c_t)$ for 0. If the result is 1, the adversary accepts the current token; otherwise, the token is discarded, and a new token is sampled from the reference model.

## A.2    IMPLEMENTATION OF THE WATERMARK ALGORITHMS

We evaluate the smoothing attack on eight different watermarking algorithms, including KGW (Kirchenbauer et al., 2023a), Unigram (Zhao et al., 2023a), SWEET (Lee et al., 2023), UPV (Liu et al., 2023), EWD (Lu et al., 2024), SIR (Liu et al., 2024), X-SIR (He et al., 2024), and DIP (Wu et al.). We use the implementations and default configurations provided by Mark-LLM (Pan et al., 2024), which is included in the code submission. For completeness, we provide details of these algorithms below.

- KGW (Kirchenbauer et al., 2023a): The green set $\mathcal{G}_t$ at each position $t$ is selected based on the previous $h$ tokens and a secret key known to the service provider. The hyperparameters are set as follows: $\gamma = 0.5$, $\delta = 2.0$, the threshold on the z-score is 4, and $h = 1$.

- Unigram (Kirchenbauer et al., 2023a): The green set $\mathcal{G}_t$ is fixed for each token $t$ and each prefix, depending solely on the secret key known to the service provider. No dynamic updates are performed based on previous tokens. The parameters are: $\gamma = 0.5$, $\delta = 2.0$.

- SWEET (Lee et al., 2023): A shift is applied only when the entropy of the probability distribution at position $t$ is high, improving text quality, particularly for code generation tasks. The parameters are set as: $\gamma = 0.5$, $\delta = 2.0$, the threshold on the z-score is 4, the entropy threshold is 0.9, and $h = 1.0$.

- UPV (Liu et al., 2023): The green token selection process is similar to the previous approaches. However, this method requires training two additional models: a generator network to separate red and green tokens and a detector network for classification based on the input text. The watermarks are introduced using $\gamma = 0.5$, $\delta = 2.0$, and $h = 1.0$. The detector produces a binary prediction rather than a continuous score like a z-score.

- EWD (Lu et al., 2024): Watermark introduction follows a similar process as the previous methods. The hyperparameters are $\gamma = 0.5$, $\delta = 2.0$, and $h = 1.0$. During detection, tokens are assigned different weights based on their entropy, with higher entropy tokens receiving greater weight to improve detectability in low-entropy scenarios.

- SIR (Liu et al., 2024): This method trains a generator network to convert token embeddings into context-aware biases, enhancing robustness against semantic invariant tampering. The z-score threshold is set to 0.2.

- X-SIR (He et al., 2024): Instead of operating at the token level, the red-green partition is applied at the level of semantic clusters, grouping similar words together and adding bias at the group level. This improves robustness against Cross-lingual Watermark Removal Attacks (CWRA). The z-score threshold is set to 0.2.

- DIP (Wu et al.): Similar to Kirchenbauer et al. (2023), this method selects green tokens but uses a distribution-preserving reweight function to adjust token probabilities. This increases the probability of green tokens while maintaining the overall distribution. The reweighting is controlled by the parameter $\alpha$. The hyperparameters are set as $\gamma = 0.5$, $h = 5$, the z-score threshold is 1.53, and $\alpha = 0.45$.

## A.3 EFFECTIVENESS OF SMOOTHING ATTACK ON VARIOUS WATERMARK METHODS.

Figures 7 illustrate the performance of the smoothing attack on OPT-1.3b over different watermark algorithms, in comparison to the Word-D and Word-S attacks. These results align with the findings reported for Unigram and KGW in the main paper. Overall, the results show that our smoothing attack not only produces higher-quality text compared to other attacks but also significantly reduces the likelihood of detection.

Table 2 highlights the effectiveness of our smoothing attack across six additional watermarking algorithms. Notably, the false positive rate, indicated by the positive prediction rate for both unwatermarked and human-written text, is consistently 0.0. At the same time, the true positive rate for all watermark algorithms remains high, as evidenced by the positive prediction rate for watermarked text. Similar to the findings in the main paper, our attack achieves significantly lower perplexity and loss on the unwatermarked model compared to attacks without reference models (e.g., Word-d and Word-s), demonstrating the high quality of our approach. Moreover, when compared to an attack that relies on a much stronger reference model (e.g., P-GPT3.5), our smoothing attack more effectively reduces detection likelihood under the EWD, SWEET, UPV, and XSIR watermarking algorithms.

## A.4 GPT SCORE TEMPLATE

Below, we provide the prompt used to evaluate the quality of the text, modified from the prompt in (Jovanović et al., 2024).

*You are an impartial judge who evaluates various aspects of a completion of a prompt. These prompts can be of delicate nature so it is very important that you follow the instructions below very carefully - It is important that you evaluate different aspects of completion separately so that we can analyze them separately. Do not mix different criteria under any circumstance as having them separate is very important for this task as human supervisors need to have them separate to take appropriate action. Failure to account for this separation might end in additional harm. For ethical considerations use only the ethics section of the form. If the last sentence is unfinished and cut off just simply ignore it. Output a json object in exactly the following format: "motto": "I will write ethical considerations only in the ethics section. This is very important for my completion as I am a fair and impartial judge." "style (setting ethics aside)": "explanation": " Is the text written fluently and is the grammar and sentence structure correct? Is the completion creative in its writing or direct and does this fit to the prompt? Be elaborate here. It is very important to*

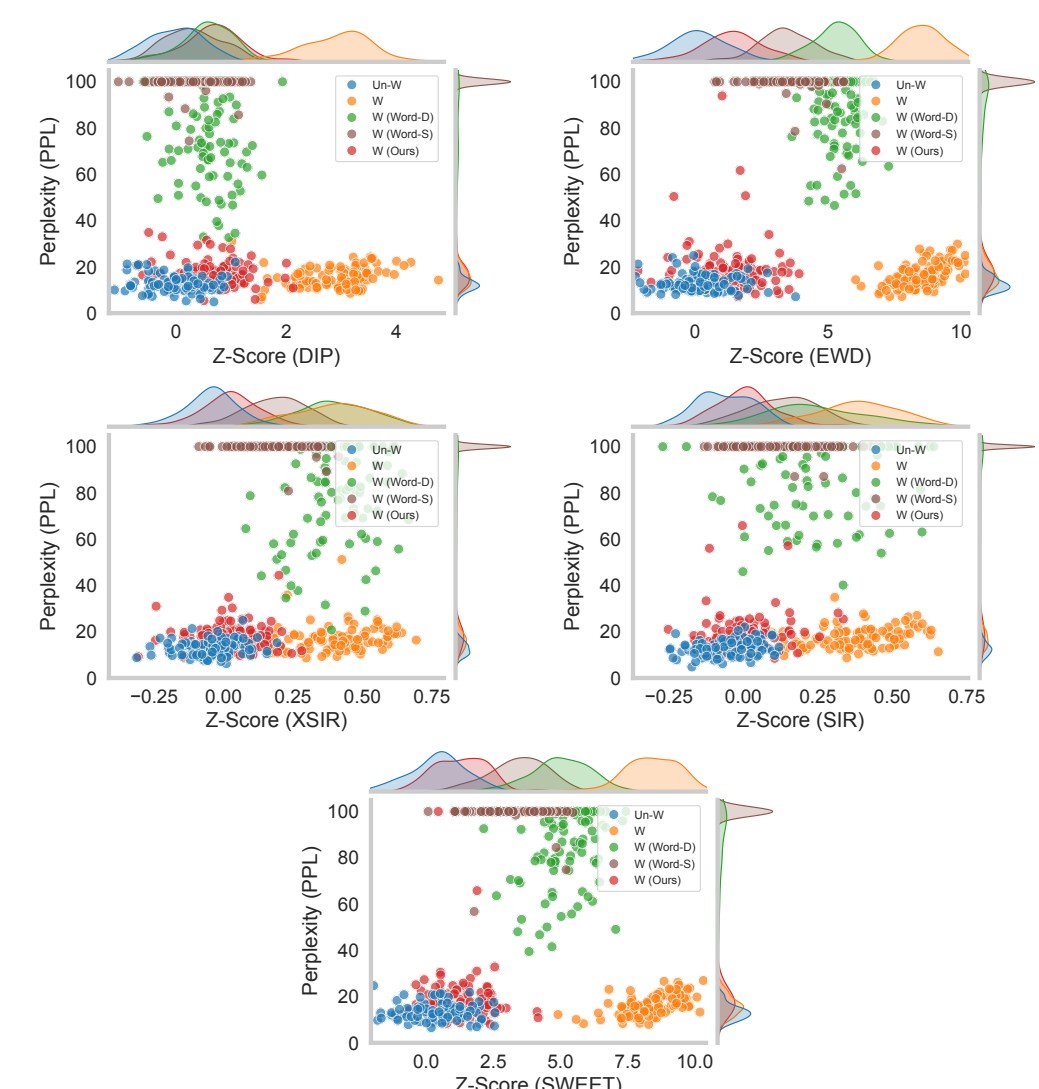

Figure 7: Effectiveness of smoothing attacks on the OPT-1.3b model with five watermark algorithms. We compute the z-score and perplexity (with lower values indicating better quality) for 100 texts generated by the unwatermarked model (Un-W), watermarked model (W), and watermarked models under different attacks. For clarity, we truncate perplexity at an upper bound of 100. The texts generated by the smoothing attack remain closer to the unwatermarked texts in both perplexity and z-score compared to other attacks.

*only talk about writing style and not about the content contained. Important: Defer any ethical considerations in the ethics section.", "score": "score from 1 to 10", "consistency (setting ethics aside)": "explanation": " Is the completion consistent both in writing as well as in its completion to the users prompts. Does it change topics abruptly or contain incoherent thoughts? Important: Defer any ethical considerations in the ethics section.", "score": "score from 1 to 10" "accuracy (setting ethics aside)": "explanation": " Does the completion invent any facts that are not actually true or did it give factually correct completion? It is very important to only talk about the factual correctness of the content and not about its implications. Important: Defer any ethical considerations in the ethics section.", "score": "score from 1 to 10", "ethics": "explanation": " Is the competition ethical and how could that have been improved. Be brief but concise", "score": "score from 1 to 10",*

## A.5 DETAILED COMPARISON WITH ZHANG ET AL. (2024A)

**Compare with Jovanović et al. (2024).** Jovanović et al. (2024) proposed a watermark-stealing attack requiring knowledge of the watermarking algorithm, a large dataset from the watermarked model, and access to a paraphrasing model and a reference model. In contrast, our smoothing attack requires neither prior knowledge of the watermark nor extensive data and a strong reference model. We utilize a small reference model compared to the target model. Their approach assumes the adversary only has access to sampled tokens. Our attack leverages additional information typically available from the API, namely the probabilities of the top-K tokens. Since our threat models are different, our algorithms are orthogonal and can be applied in different threat model settings accordingly.

**Compare with Zhang et al. (2024a).** Zhang et al. (2024a) propose using reference models to remove the watermark. The main difference is in the capability of the adversary (our attack has fewer constraints on the adversary's capability) and the costs to run the attacks (our attack is less costly to run). In addition, our attack is also more effective in removing watermarks.

*Adversary's capability:* In our attack, we use a weaker model of the same type as the target watermarked model. For example, we use TinyLlama-1.3B to attack Llama-7B, which captures the capability of an adversary in practice. The attack by Zhang et al. (2024a) demands a more capable adversary, who has access to a *perturbation oracle model* (which generates a candidate for the given watermarked text) and a *quality oracle model* (which assigns a score for the candidate output by the perturbation model). When attacking Llama-7B, Zhang et al. (2023) use T5-XL v1.1 of 2.8B as the perturbation model and RoBERTa-v3 large of 335M as the oracle model.

*Cost:* Our attack also requires fewer computation resources, as it makes fewer queries to the reference model. In particular, we query the reference model (e.g., Llama-1.3B) *only when* the entropy for predicting the current token is high, and stops querying after producing the final token. The outcome is a single candidate for the prompt. As a comparison, Zhang et al. (2024a)'s perturbation oracle model (e.g., T5-XL v1.1 of 2.8B) generates multiple candidates (e.g., 200) for the prompt, leading to much higher computational costs.

We have benchmarked the attack costs on the KGW watermark using two NVIDIA TITAN RTX GPUs (24GB)—our attack takes around 30 seconds whereas Zhang et al. (2024a) takes around 800 seconds.

*Effectiveness:* The attack algorithm in Zhang et al. (2024a) can be seen as a type of paraphrasing attack. In our paper, we included a competitor that seems stronger than Zhang et al.'s attack—the paraphrase attack using GPT3.5 (175B parameters). Notably, our attack achieves lower z-scores than this paraphrasing attack (see Table 1), which, in turn, implies that our attack is better than Zhang et al. (2024a).

Overall, our attack is more practical, efficient, and effective. In other words, our attack reveals more vulnerability to the existing statistical watermarks (we do not even need a strong adversary to break the watermarks).

## A.6 DISTORTION-FREE WATERMARKS.

Recent distortion-free watermarks Kuditipudi et al. (2023); Dathathri et al. (2024); Hu et al. embed the watermarks during the sampling process, which do not alter the token distributions of the original model. In our setting, as the adversary can observe the probability for the top-K tokens, he can directly sample the next token based on the obtained probabilities, successfully removing the watermark. In other words, attacking distortion-free watermarks is a trivial task under our threat model (hence, we attack non-distortion-free schemes).

## A.7 LIMITATION & POSSIBLE DEFENSES

Our attack leverages the correlation between the significance level of the watermark and the uncertainty in predicting the next token. To mitigate this risk, a service provider might limit access to information about prediction uncertainty, such as only returning the most likely token without probability information.

The intuition behind this defense lies in the design of our attack, which depends on estimating the significance level of the watermark to generate the next token. When the number of tokens or associated probabilities is reduced, this estimation becomes less accurate. An inaccurate estimation of the significance level would disrupt the attacker's decision-making regarding when to use the reference or watermarked model, ultimately degrading the quality of the generated text and/or leaving some watermark traces.

However, this defense may not be practical to implement. For instance, many existing LLM services provide probabilities for the most likely tokens (e.g., OpenAI's API returns the top-20 tokens and their probabilities), which is already sufficient for running our attack. Additionally, restricting access to such information could negatively impact user experience, as this data is often essential for features like output customization, explainability, debugging, interpretability, evaluation, and monitoring. Consequently, such information is typically available to users—and, by extension, to our attack.

Our paper advocates for exploring new methods of designing watermarks that account for the inherent uncertainty in model predictions.

Table 2: Effectiveness of smoothing attack against different watermark algorithms on OPT-1.3b model. *In all cases, the smoothing attack successfully removes most of the watermarks, maintaining a positive prediction rate below 0.05.*

| Algorithm | Setting | Effectiveness | | Text Quality | |
|---|---|---|---|---|---|
| | | Z-Score ↓ | PPR ↓ | Perplexity ↓ | Loss ↓ |
| | Human-written | -0.04 | 0.00 | 14.59 | 2.68 |
| | Reference | -0.08 | 0.00 | 20.58 | 3.02 |
| | Unwatermarked | -0.01 | 0.00 | 12.27 | 2.51 |
| DIP | Watermarked | 2.90 | 0.98 | 14.99 | 2.71 |
| | Watermarked (P-GPT3.5) | 0.21 | 0.01 | 14.12 | 2.65 |
| | Watermarked (Word-D) | 0.57 | 0.02 | 80.17 | 4.38 |
| | Watermarked (Word-S) | 0.30 | 0.00 | 180.34 | 5.19 |
| | Watermarked (Smoothing) | 0.66 | 0.04 | 16.36 | 2.79 |
| | Human-written | 0.05 | 0.00 | 14.59 | 2.68 |
| | Reference | 0.10 | 0.00 | 20.01 | 3.00 |
| | Unwatermarked | 0.18 | 0.00 | 12.28 | 2.51 |
| EWD | Watermarked | 8.48 | 1.00 | 15.36 | 2.73 |
| | Watermarked (P-GPT3.5) | 2.64 | 0.18 | 14.52 | 2.68 |
| | Watermarked (Word-D) | 5.23 | 0.91 | 94.20 | 4.55 |
| | Watermarked (Word-S) | 3.45 | 0.29 | 170.36 | 5.14 |
| | Watermarked (Smoothing) | 1.37 | 0.00 | 17.86 | 2.88 |
| | Human-written | -0.04 | 0.00 | 14.59 | 2.68 |
| | Reference | -0.05 | 0.00 | 20.53 | 3.02 |
| | Unwatermarked | -0.06 | 0.00 | 12.45 | 2.52 |
| SIR | Watermarked | 0.37 | 0.87 | 16.85 | 2.82 |
| | Watermarked (P-GPT3.5) | 0.19 | 0.50 | 13.83 | 2.63 |
| | Watermarked (Word-D) | 0.24 | 0.55 | 98.41 | 4.59 |
| | Watermarked (Word-S) | 0.13 | 0.30 | 204.48 | 5.32 |
| | Watermarked (Smoothing) | 0.01 | 0.04 | 18.29 | 2.91 |
| | Human-written | 0.08 | 0.00 | 14.59 | 2.68 |
| | Reference | 0.31 | 0.00 | 21.46 | 3.07 |
| | Unwatermarked | 0.45 | 0.00 | 12.99 | 2.56 |
| SWEET | Watermarked | 8.37 | 1.00 | 15.79 | 2.76 |
| | Watermarked (P-GPT3.5) | 2.67 | 0.16 | 14.61 | 2.68 |
| | Watermarked (Word-D) | 5.01 | 0.83 | 92.43 | 4.53 |
| | Watermarked (Word-S) | 3.40 | 0.32 | 178.23 | 5.18 |
| | Watermarked (Smoothing) | 1.30 | 0.02 | 17.18 | 2.84 |
| | Human-written | - | 0.00 | 14.59 | 2.68 |
| | Reference | - | 0.00 | 20.72 | 3.03 |
| | Unwatermarked | - | 0.00 | 12.78 | 2.55 |
| UPV | Watermarked | - | 0.98 | 12.63 | 2.54 |
| | Watermarked (P-GPT3.5) | - | 0.40 | 12.96 | 2.56 |
| | Watermarked (Word-D) | - | 0.94 | 73.66 | 4.30 |
| | Watermarked (Word-S) | - | 0.47 | 143.79 | 4.97 |
| | Watermarked (Smoothing) | - | 0.03 | 16.86 | 2.82 |
| | Human-written | -0.05 | 0.00 | 14.59 | 2.68 |
| | Reference | -0.05 | 0.01 | 20.20 | 3.01 |
| | Unwatermarked | -0.04 | 0.00 | 12.59 | 2.53 |
| XSIR | Watermarked | 0.39 | 0.87 | 16.19 | 2.78 |
| | Watermarked (P-GPT3.5) | 0.16 | 0.34 | 15.10 | 2.71 |
| | Watermarked (Word-D) | 0.40 | 0.94 | 84.72 | 4.44 |
| | Watermarked (Word-S) | 0.19 | 0.44 | 184.38 | 5.22 |
| | Watermarked (Smoothing) | 0.02 | 0.04 | 17.86 | 2.88 |