# OpenReview forum: "Watermark Smoothing Attacks against Language Models"
_ICLR.cc/2025/Conference — Submitted to ICLR 2025_

### Official Review · Reviewer_NfPb · 2024-10-29

**Soundness:** 2
**Presentation:** 2
**Contribution:** 2
**Rating:** 3
**Confidence:** 4

**Summary:**

The paper introduces a "smoothing attack" that bypasses statistical watermarking in large language models (LLMs). By blending outputs from the watermarked model with a weaker reference model, it removes watermarks without impacting text quality on PPL.

**Strengths:**

1. The writing is easy to follow.

2. Propose a smoothing attack scheme against statistical watermarking, and show that the significance level $S_t$ is highly correlated with the total variation distance.

**Weaknesses:**

1. The applicability of this method is limited, as obtaining a high-quality reference model is often not possible (e.g. for GPT-4). Additionally, it requires access to token logits, meaning it is not a purely black-box approach as claimed.

2. In line 146. The authors overclaim that their attack is universally applicable to all statistical watermarking schemes. However, many watermarking schemes [1,2,3] do not use a green list, and their proposed method cannot be applied.

3. Additional metrics are needed to better reflect the quality of the generated text. PPL tends to favor a distribution similar to that of the oracle model, which can introduce bias. It would be more informative to include straightforward metrics, such as BLEU in machine translation, to provide a clearer evaluation.

4. The paper lacks key baseline results needed to demonstrate the effectiveness of the proposed method.  Naive smoothing using $\lambda \tilde{P}(x)+(1-\lambda) P^{ref}(x)$ can also remove the watermark while preserving part of the text quality.

5. The choice of z-score threshold used in the experiments is unclear. It would be more straightforward to present the true positive rates at specific theoretical false positive rates, providing a clearer understanding of the method’s performance.

6. The experimental settings for certain tests are suboptimal. For instance, in Table 2, the z-score for XSIR and SIR is too low, indicating that the watermark strength in the original watermarked model is insufficient.

[1] Kuditipudi, R., Thickstun, J., Hashimoto, T. and Liang, P., 2023. Robust distortion-free watermarks for language models. arXiv preprint arXiv:2307.15593.

[2] Hu, Z., Chen, L., Wu, X., Wu, Y., Zhang, H. and Huang, H., 2023. Unbiased watermark for large language models. arXiv preprint arXiv:2310.10669.

[3] Dathathri, S., See, A., Ghaisas, S., Huang, P.S., McAdam, R., Welbl, J., Bachani, V., Kaskasoli, A., Stanforth, R., Matejovicova, T. and Hayes, J., 2024. Scalable watermarking for identifying large language model outputs. Nature, 634(8035), pp.818-823.

**Questions:**

1. Why in Figure 1, top-p sampling (right figure) has some points with the total variation distance being 0 or 1, but top-k sampling (middle figure) does not?
2. How many queries (prefixes) do you use for computing the bin index as described in Lines[261-266]?

---

> ### Author Response · Authors · 2024-11-22
> **Response to Reviewer NfPb**
>
> **W1 (threat model):**
> > The applicability of this method is limited, as obtaining a high-quality reference model is often not possible (e.g. for GPT-4). Additionally, it requires access to token logits, meaning it is not a purely black-box approach as claimed.
>
> **Response:** We believe there is a misunderstanding.
>
> First, our attack relies on reference models that are much *weaker* than the target watermarked model. For example, when attacking Llama2-7b, we use the TinyLlama-1.3b as the reference model. The main message of our paper is that we do not need a strong adversary to break the existing statistical watermarks, which are validated by the experiment results.
>
> Second, our attack relies on *probabilities from the top-K tokens* of the watermarked model rather than direct access to token logits. This information is typically accessible in black-box scenarios, such as through the OpenAI’s API (with k=20).
>
> Therefore, our proposed method is in general applicable, for attackers with limited capabilities aiming at black-box models.
>
> ---
>
> **W2 (distortion-free watermarking schemes):**
> > In line 146. The authors overclaim that their attack is universally applicable to all statistical watermarking schemes. However, many watermarking schemes [1,2,3] do not use a green list, and their proposed method cannot be applied.
>
> **Response:** Thanks for pointing this out.  We are aware of distortion-free watermarking schemes, which *do not alter* the token distributions of the original model. Those algorithms introduce the watermarks during the sampling process without affecting the probability distribution, the mapping from the probability distribution to the token, without changing the probability distribution.  In our setting, as the adversary can observe the probability for the top-K tokens (which is a valid assumption; see the above example of OpenAI), he can directly sample the next token based on the obtained probabilities, successfully removing the watermark.
>
> In other words, attacking distortion-free watermarks is a trivial task under our threat model (hence, we attack non-distortion-free schemes). We have added a clarification to this issue and adjusted our claims accordingly in the revised version (see Appendix A.5).
>
> **W3 (quality metric):**
> > Additional metrics are needed to better reflect the quality of the generated text. PPL tends to favor a distribution similar to that of the oracle model, which can introduce bias. It would be more informative to include straightforward metrics, such as BLEU in machine translation, to provide a clearer evaluation.
>
> **Response:** We have added the new results in response to this question. Please refer to our [general response](https://openreview.net/forum?id=1AYrzmDK4V&noteId=wzjDTw8cbD).
>
> ---
>
> **W4 (naive smoothing baseline):**
> > • The paper lacks key baseline results needed to demonstrate the effectiveness of the proposed method. Naive smoothing using  $\lambda \tilde{P}(x) + (1- \lambda)P^{ref}(x)$ can also remove the watermark while
> preserving part of the text quality.
>
> **Response:** We appreciate the reviewer’s intuition. The suggested attack indeed was considered once as a candidate, but there are several caveats to running this attack in practice.
>
> 1. It is difficult to obtain $P(x)$ of the target model. In our setting, the attacker only has access to the top-K probabilities and the corresponding tokens, but not the whole probability distribution.
> 2. Even if 1. is possible, the combination of the probability distributions of the target model and the reference model together may not be a valid probability distribution in the first place, since they could be defined on different spaces (i.e., the tokenizers may be different). Our solution avoids this issue by either sampling from the target model or the reference model.

---

> ### Author Response · Authors · 2024-11-22
> **Response to Reviewer NfPb -- Continued**
>
> ---
>
> **W5 (z-score threshold and TPR)**:
> > The choice of z-score threshold used in the experiments is unclear. It would be more straightforward to present the true positive rates at specific theoretical false positive rates, providing a clearer understanding of the method’s performance.
>
> **Response:** Sorry for the confusion. The z-score thresholds used in our experiments are presented in Section 4.1 and detailed in Appendix A2 (see page 14).
>
> For clearer presentations, we have also reported the Positive Prediction Rate (PPR), which measures the fraction of test inputs that are predicted as watermarked by the watermark detector. From Tables 1 (page 8) and 2 (page 17), it is clear that our attack consistently outperforms almost all other attacks by achieving lower PPRs (except sometimes outperformed by the impractical paraphrasing attack with GPT4).
>
> In what follows, we provide a detailed explanation of why lower PPRs are better.
>
> There are two types of texts, positive samples (those generated from the watermarked model with/without attacks) and negative samples (those written by humans). Considering positive samples (i.e., texts generated from the watermarked model with/without attacks), PPR is computed as the fraction of these positive samples correctly identified as positive/watermarked by the detector, which reflects the *True Positive Rate*. For negative samples (text generated from unwatermarked models or human written text), the PPR is computed as the fraction of these negative samples misidentified as positive/watermarked by the detector, which reflects the *False Positive Rate*. Hence, a lower PPR indicates a stronger attack against watermark models.
>
> We have added the above clarification in the revised version (Section 4).
>
> ---
>
> **W6 (setups for XSIR and SIR):**
> > The experimental settings for certain tests are suboptimal. For instance, in Table 2, the z-score for XSIR and SIR is too low, indicating that the watermark strength in the original watermarked model is insufficient.
>
> **Response:** We believe this is a misunderstanding. Indeed, it is not sufficient to focus on the z-score metric alone (also see the above response). That is exactly why we have also included the PPR as a metric.
>
> For the XSIR and SIR methods, their PPRs are both 0.87, meaning that the watermarks are detectable. Hence, this setup is not suboptimal. Using our attack, we managed to reduce the PPR to 0.04 (see Table 2 on page 17). Hence, our attack is effective.
>
> ---
>
> **Q1 (total variation distance):**
> > Why in Figure 1, top-p sampling (right figure) has some points with the total variation distance being 0 or 1, but top-k sampling (middle figure) does not?
>
> **Response:** This is a good point. This difference is attributed to the different natures of top-p and top-k sampling.
>
> - In top-p sampling, we sample from the smallest set of tokens whose sum of probabilities is at least p (we set p to 0.8). Hence, it is likely that there are only a few candidates to sample from the watermarked model and the unwatermarked model, making the sampling process almost *deterministic*. As a result, the total variation distance (TVD) between the empirical distributions of the watermarked and unwatermarked models is **i**): either very large or becomes 1 (when the two candidate sets have no overlap at all); or **ii**): very small or becomes 0 (when the two candidate sets are almost identical, or both containing the same one candidate).
> - In top-k sampling, we sample from the tokens of the top-K probabilities. When we set k to 20, there are more candidates to sample from, making the sampling process more stochastic/less deterministic. As a result, the above scenarios **i**) and **ii**) are not likely to happen.
>
> Despite the above difference, both sampling methods reflect the positive correlation between the significance level and the TVD between the token distributions of the watermarked and unwatermarked models. We can then exploit this correlation to design the attack.
>
> ---
>
> **Q2 (number of prefixes):**
> > How many queries (prefixes) do you use for computing the bin index as described in Lines[261-266]?
>
>
> **Response:** Thanks for asking. 200 prefixes are used to compute the upper and lower bounds for constructing the bins. Note that all 200 prefixes are obtained as **one response** output by the target watermarked model, and the upper and lower bounds can be reused by our attack on different texts. Hence, **this cost is negligible**.

---

### Official Review · Reviewer_RzfD · 2024-10-30

**Soundness:** 2
**Presentation:** 3
**Contribution:** 2
**Rating:** 3
**Confidence:** 3

**Summary:**

This work develops a smooth attack in the “green-red list” watermarking framework. The paper shows that a smooth attack makes it easier to bypass the detector while still preserving the quality of the text.

**Strengths:**

Many existing methods for statistical watermarking have primarily concentrated on the generation and detection of watermarks. This paper takes a different approach by examining statistical watermarking from a new perspective. This perspective is interesting and may also aid in the development of improved watermark generation and detection techniques.

**Weaknesses:**

1. The significance level $S_t$ is unobserved and was estimated using a surrogate quantity, $c_t$. Though the authors showed that there is generally a negative correlation between $c_t$ and $S_t$, this is only a weak justification. It is possible that a small $c_t$ would correspond to a large $S_t$ in some situations, e.g., when $K$ is small.
2. The method only applies to the “green-red list” watermarking scheme, which is known to be biased because it does not preserve the original text distribution. In contrast, there are unbiased watermarking methods (e.g., Kuditipudi et al., 2023; Aaronson, 2023). It is unclear if the proposed method applies to unbiased watermarking schemes. Perhaps the authors can provide more discussions about how their method might be adapted or extended to work with unbiased watermarking schemes.
3. The paper lacks a rigorous theoretical analysis of the effect of the smooth attack on the text quality, e.g., bounds on how much the smoothing attack can affect certain text quality metrics.

**Questions:**

1. In Table 1, Watermark (smoothing) has a lower perplexity than Watermark (or even Unwatermark) in some cases (e.g., Llama2-7b). In other words, the attack can even improve the quality of the text, which seems counterintuitive as the reference model is weaker. This also raises a concern about whether perplexity is the right measure to look at the quality of a text here. The authors may want to include other text quality metrics in the numerical studies.
2. I would like to know if the authors can discuss the potential pitfalls of their methods, e.g., provide concrete examples or scenarios where their smooth attack might fail, and discuss the implications of such failures

---

> ### Author Response · Authors · 2024-11-22
> **Response to Reviewer RzfD**
>
> **W1 (weaker correlation when k is small):**
> > The significance level St is unobserved and was estimated using a surrogate quantity, ct. Though the authors showed that there is generally a negative correlation between cy and St, this is only a weak justification. It is possible that a small c would correspond to a large St in some situations, e.g., when K is small.
>
> **Response:**
> We appreciate the reviewer for raising this concern.
> Indeed, the significance level was estimated based on a surrogate quantity and there could be errors in this estimation. In our design, we circumvent this issue by resorting to a *soft assignment of token distributions:* when the estimated significance level is high, it is *more likely (but not deterministic)* that we sample from the watermark-free reference model, and vice versa. Perhaps due to this design, our method does not suffer much from the aforementioned estimation error and performs the best among existing competitors (except for the unrealistic paraphrase attack using GPT4). We will leave further improvement as a future work direction.
>
> ---
>
> **W2 (distortion-free watermarking schemes):**
> > The method only applies to the “green-red list” watermarking scheme, which is known to be biased because it does not preserve the original text distribution. In contrast, there are unbiased watermarking methods (e.g., Kuditipudi et al., 2023; Aaronson, 2023). It is unclear if the proposed method applies to unbiased watermarking schemes. Perhaps the authors can provide more discussions about how their method might be adapted or extended to work with unbiased watermarking schemes.
>
>
> **Response:**
> Kuditipudi et al., 2023; and Aaronson, 2023 do not alter the token distributions of the original model. Such watermark schemes are trivial to attack under our threat model, where the adversary can observe the top-K probabilities and the corresponding tokens (this information is in general accessible to users, e.g., via OpenAI’s API). As the adversary can observe the unaltered probabilities, effectively, he can directly sample tokens from the unwatermarked model, obtaining the unwatermarked output (this problem is trivial). Hence, we focus on watermark schemes that do not preserve the original probability distributions. We have added a clarification to this issue and adjusted our claims accordingly in the revised version (see Appendix A.5).
>
> ---
>
> **W3 (theoretical analysis of text quality):**
> > The paper lacks a rigorous theoretical analysis of the effect of the smooth attack on the text quality, e.g., bounds on how much the smoothing attack can affect certain text quality metrics.
>
>
> **Response:**
> We acknowledge the reviewer’s concern, but this does not demerit the contribution of our work.
>
> Our work is not focused on theoretical analysis; rather, it provides a practical attack that is good enough to demonstrate the limitations of the current ``Green-red list'' statistical watermarking schemes, calling for new watermark techniques.
>
> Besides, sometimes our attack generates texts of higher quality than the target watermarked model, especially when the distortion caused by watermarks is significant (See Figure 5 in the paper).
>
> ---
>
> **Q1 (About quality metric):**
> > In Table 1, Watermark (smoothing) has a lower perplexity than Watermark (or even Unwatermark) in some cases (e.g., Llama2-7b). In other words, the attack can even improve the quality of the text, which seems counterintuitive as the reference model is weaker. This also raises a concern about whether perplexity is the right measure to look at the quality of a text here. The authors may want to include other text quality metrics in the numerical studies.
>
> **Response:**
> We have added the new results in response to this question. Please refer to our [general response](https://openreview.net/forum?id=1AYrzmDK4V&noteId=wzjDTw8cbD).
>
> ---
>
> **Q2 (potential pitfalls):**
> > I would like to know if the authors can discuss the potential pitfalls of their methods, e.g., provide concrete examples or scenarios where their smooth attack might fail, and discuss the implications of such failures
>
> **Response:**
> Our attack may encounter challenges in scenarios where the adversary has access to only a limited number of token probabilities (i.e., reducing the value of K), causing large errors in the estimate of the significance level (as the reviewer pointed out in the first question).  However, as we have mentioned, our attack circumvents this pitfall by using a soft assignment of token distributions, and performs considerably well under most experimental setups (we set K=20, which seems to be a valid assumption so far).  We have included a discussion on this limitation in the revised version (see Appendix A.7).
>
> Please also refer to our response for the [potential defense](https://openreview.net/forum?id=1AYrzmDK4V&noteId=TdIMt2V76X)

---

> > ### Comment · Reviewer_RzfD · 2024-12-02
> >
> > I thank the authors for responding to my questions; however, some of my concerns remain unresolved. Firstly, the authors have not provided a convincing conceptual (W1) and theoretical (W3) justification for their scheme. Second, it is difficult to make a fair comparison of the quality of the texts produced by different methods. The newly added results based on GPT-4 would clearly favor texts similar to those produced by GPT-4 (but does it really mean the text has a higher quality?) Lastly, the fact that the method is only applicable to the "green-red list" watermarking scheme limits its overall scope.

---

> > > ### Author Response · Authors · 2024-12-03
> > > **Response to Reviewer RzfD**
> > >
> > > We appreciate the reviewer's feedback. Below, we address each concern in detail:
> > > - **Theoretical Justification (W3):**  We acknowledge the reviewer's request for additional theoretical justification. However, given the complexity and black-box nature of large language models (LLMs), we believe that the most convincing evidence for the efficacy of our attack is its empirical success.
> > > - **Fair Comparison of Text Quality:**  We recognize the broader challenges in defining and comparing text quality across different methods, a fundamental issue within the field of LLMs that extends beyond watermarking. In our revised version, we have incorporated two reasonable and widely used metrics, perplexity and LLM-as-a-Judge, to evaluate text quality.
> > >
> > > - **Applicability Beyond the "Green-Red List" Watermarking Scheme:**  Under our threat model, other watermarking approaches can be bypassed by sampling from the top-k probabilities, which not only avoids watermarked outputs but also preserves text quality. This is precisely why our evaluation focuses on the green-red list method.

---

### Official Review · Reviewer_a4FL · 2024-11-01

**Soundness:** 2
**Presentation:** 3
**Contribution:** 3
**Rating:** 6
**Confidence:** 3

**Summary:**

In this paper, the authors introduce a novel watermark-removal attack that requires only a small watermark-free reference model. The attacker first estimates the probability of the generated token at position i being in the watermark's green list, which correlates with the relative confidence of the most likely token among the top k tokens. According to the confidence score, the attacker then combines the probability distributions at position i from both the watermarked model and the reference model to sample the token. This approach effectively evades watermark detection while maintaining high text quality.

**Strengths:**

- I find the proposed method very interesting and quite different from the previous work. Meanwhile, the method doesn't require a strong oracle model like a paraphrasing attack, which makes the threat model more realistic.
- I really enjoy reading this paper, especially section 3.1, which gives readers a lot of insights.
- The results look positive and a lot of different watermarking schemes are covered (most results are presented in the appendix).

**Weaknesses:**

- The proposed method relies on using the logits/output probabilities of the watermarked model. This might limit the attack to some API models that may not return the logits/probabilities or only return top-k probabilities or even calibrated probabilities.
- The paper uses perplexity or loss to measure the text quality, but I think it's not enough to show the quality of the text. For example, the model can generate an answer for a math question with a very low perplexity, but the answer is completely wrong. So, I think it will be more helpful if the authors can include more text quality metrics like P-SP used in [1] or even a model-based evaluation like asking a large oracle model which generation is preferable.
- I think it's also helpful to the paper if the answers can show the results under different data distributions instead of overall c4.

[1] Kirchenbauer, J., Geiping, J., Wen, Y., Shu, M., Saifullah, K., Kong, K., Fernando, K., Saha, A., Goldblum, M., & Goldstein, T. (2023). On the Reliability of Watermarks for Large Language Models. ArXiv, abs/2306.04634.

**Questions:**

- Can the authors provide a baseline that uses the local reference model to do the paraphrase attack?
- What could be potential adaptive defenses for this attack?

---

> ### Author Response · Authors · 2024-11-22
> **Response to Reviewer a4FL**
>
> **W1 (threat model):**
> >The proposed method relies on using the logits/output probabilities of the watermarked model. This might limit the attack to some API models that may not return the logits/probabilities or only return top-k probabilities or even calibrated probabilities.
>
> **Response:**
>
> We believe this is a misunderstanding. Our attack relies on the *top-K probabilities* from the watermarked model instead of logits or full output probabilities for all tokens. As we have mentioned above, this information is generally accessible to the attacker/user of the model. Our attack is also not affected under standard calibration techniques, as our attack uses the *relative difference* between the probabilities of the top-1 and top-K tokens for estimating the significance level.
>
> ---
>
> **W2 (quality metric):**
> > The paper uses perplexity or loss to measure the text quality, but I think it's not enough to show the quality of the text. For example, the model can generate an answer for a math question with a very low perplexity, but the answer is completely wrong. So, I think it will be more helpful if the authors can include more text quality metrics like P-SP used in [1] or even a model-based evaluation like asking a large oracle model which generation is preferable.
>
> **Response:**
>
> We have added the new results in response to this question. Please refer to our [general response](https://openreview.net/forum?id=1AYrzmDK4V&noteId=wzjDTw8cbD).
>
> ---
>
>
> **W3 (data):**
> > I think it's also helpful to the paper if the answers can show the results under different data distributions instead of overall c4.
>
> **Response:**
>
> Thank you for your insights. We agree that it is an interesting research direction to explore the effect of watermark defense/attack schemes under different data distributions. However, given the comprehensive nature of the watermark algorithms we have evaluated, we primarily focused on the commonly used benchmark datasets [1, 2, 3, 4, 5]. Further exploration in individual data domains is beyond the scope of this work.
>
> ---
>
> **Q1 (a new baseline and more experiment results):**
> > Can the authors provide a baseline that uses the local reference model to do the paraphrase attack?
>
>
> **Response:**
> Sure. The suggested baseline, according to our newly added experiments (see below), is less effective than the baseline which queries the reference model directly.
>
> In particular, we use the reference model to phrase the text generated from the watermarked model, using the prompt “Rewrite the text in the parenthesis (<WATERMARKED TEXT>):”. In the table below, we compare the results between using the reference model directly and using the reference to paraphrase the text from the watermarked OPT-1.3b.
>
> |  |  | Z-score | PPR | PPL |
> | --- | --- | --- | --- | --- |
> | KGW | Reference model  | 0.21 | 0.0 | 19.75 |
> |  | Paraphrasing using reference model | 1.671  | 0.15  | 42.109 |
> | Unigram | Reference model  | -0.07 | 0.00 | 19.51  |
> |  | Paraphrasing using reference model | 0.73  | 0.05  | 37.64 |
>
> As we can see, this baseline achieves lower quality (PPL) and higher detection rate (PPR) and z-score (easier to detect). Overall, using the reference model to paraphrase watermarked text is worse than querying the reference model directly.
>
> We would like to thank the reviewer for the instructive comments.

---

> ### Author Response · Authors · 2024-11-22
> **Response to Reviewer a4FL -- Continued**
>
> **Q2 (potential defenses)**:
> > What could be potential adaptive defenses for this attack?
>
>
> **Response:** We first recall the idea of our attack, and then present the idea of a potential defense.
>
> Idea of our attack: Our attack leverages the correlation between the significance level of the watermark and the uncertainty of predicting the next token, based on which we query either the watermarked model or the reference model for generating the text token.
>
> Idea of defense: Hence, a natural idea to defend our attack is to *limit the attacker’s access* to the information on the significance level of the watermark. The more straightforward way to do that is to make the watermarked model only return the most likely token without other information or alternatively, return only a few of the most likely tokens and the associated token probabilities.  This would cause the estimation of the *significance level of the watermark* to be less accurate, affecting the attacker’s decision on when to use the reference model or the watermarked model, ultimately affecting the quality of the generated text and/or leaving some watermark traces in it.
>
> However, this defense may ***not be practical to deploy***. As an example, some existing LLM services return the probabilities of the most likely tokens (e.g., Open AI’s API returns the top-20 probabilities and the associated tokens), which already provide enough information to run our attack. Besides, limiting a user’s access to such information may do more harm than good, as this information may be crucial for good user experience (e.g., output customization, explainability, debugging, interpretability, evaluation, and monitoring). Therefore, such information is often accessible to a user and; hence, also to our attack.
>
> Thus, the effectiveness of our attack (there does not yet exist a practical defense) exemplifies the need for new watermark techniques. We have included a discussion on this limitation in the revised version (see Appendix A.7).
>
>
> ----
> [1] Kirchenbauer, John, et al. "A watermark for large language models." ICML, 2023.
>
> [2] Zhao, Xuandong, et al. "Provable Robust Watermarking for AI-Generated Text." ICLR, 2024.
>
> [3] Liu, Aiwei, et al. "An unforgeable publicly verifiable watermark for large language models." ICLR, 2023.
>
> [4] Lu, Yijian, et al. "An Entropy-based Text Watermarking Detection Method." ACL, 2024.
>
> [5] Liu, Yepeng, and Yuheng Bu. "Adaptive Text Watermark for Large Language Models." ICML, 2024.

---

> > ### Comment · Reviewer_a4FL · 2024-11-26
> > **Thank you for your response**
> >
> > I really appreciate the authors' detailed response. Therefore, I keep my score positive.

---

### Official Review · Reviewer_f25P · 2024-11-04

**Soundness:** 3
**Presentation:** 3
**Contribution:** 1
**Rating:** 5
**Confidence:** 4

**Summary:**

The paper proposes an automatic method for editing watermarked text from a language model to evade watermark detection using another (weaker) language model. The paper mainly considers the "red-green list" watermark of Kirchenbauer et al. and variants thereof, though the techniques should presumably generalize to other watermarks.

**Strengths:**

The paper proposes a heuristic to estimate which tokens contribute the most to the overall watermark signal and removes the watermark by editing these tokens using another language model. The idea is interesting, and the paper empirically validates the effectiveness of their attack across different watermarks, language models, and datasets. These results clearly establish the effectiveness of the attack in practice.

**Weaknesses:**

The paper distinguishes its main contributions from prior work by arguing that prior work on automatically removing watermarks involved using language models that were at least as strong as the original watermarked language model. However, one notable exception is the work of Zhang et al. [1], who seem to also focus on removing watermarks using weaker language models. This work is cited in the present paper but not discussed in any detail. It would be great if the authors can update their paper with a discussion of how their work differs from [1]. Otherwise, the novelty/significance of the main contributions over prior work is not clear.


[1] Zhang et al. (2023) Watermarks in the Sand: Impossibility of Strong Watermarking for Generative Models. https://arxiv.org/abs/2311.04378

**Questions:**

What are the main differences between this work and that of Zhang et al. (2023)? (see Weaknesses section)

---

> ### Author Response · Authors · 2024-11-22
> **Comparision with Zhang et al. (2023)**
>
> > What are the main differences between this work and that of Zhang et al. (2023)? (see Weaknesses section)
>
> The main differences lie in the adversary’s capabilities and the cost of executing the attacks. Our approach imposes *fewer constraints* on the adversary’s capabilities and is *less expensive* to run. Moreover, our attack is *more effective* at removing watermarks.
>
> 1. Adversary's capability:
>     - In our attack, we use a weaker model of the same type as the target watermarked model, which captures the capability of an adversary in practice. For example, we use TinyLlama-1.3b to attack Llama2-7b.
>     - The attack by Zhang et al. (2023) demands a more capable adversary, who has access to a **perturbation oracle model** (which generates a candidate for the given watermarked text) and a **quality oracle model** (which assigns a score for the candidate output by the perturbation model). When attacking Llama2-7b, Zhang et al. (2023) use T5-XL v1.1 of 2.8b as the perturbation model and RoBERTa-v3 large of 335M as the oracle model. Overall, their adversary is stronger than ours.
> 2. Cost:
>     1. Our attack also requires fewer computation resources, as it makes fewer queries to the reference model. In particular, we query the reference model (e.g., TinyLlama-1.3b) *only when* the entropy for predicting the current token is high, and stops querying after producing the final token. The outcome is a *single candidate* for the prompt.
>     2. As a comparison, Zhang et al. ‘s perturbation oracle model (e.g., T5-XL v1.1 of 2.8b) generates *multiple candidates* (e.g., 200 candidates) for one prompt, leading to much larger computational costs.
>     3. We have benchmarked the attack costs on the KGW watermark using two NVIDIA TITAN RTX GPUs (24GB)—our attack takes around 30 seconds whereas Zhang et al.’s takes around 800 seconds (under the default setting in their paper).
> 3. Effectiveness:
>     1. We have run some preliminary experiments to confirm this—our attack achieves a z-score of -0.0731, while Zhang et al.’s only achieves a z-score of 0.3747 (a smaller z-score indicates better performance in watermark removal). Hence, our attack is more effective under this metric. This comparison is based on the KGW watermarks on the OPT-2.7b model.
>     2. In our original draft, we have already included a competitor that seems stronger than Zhang et al.’s attack—the paraphrase attack using GPT-3.5 (175B parameters). Notably, our attack achieves lower z-scores than this paraphrasing attack (see page 8 table 1), which, in turn, implies that our attack is better than Zhang et al.’s in removing the watermarks.
>
> Overall, our attack is more practical, efficient, and effective. In other words, our attack reveals more vulnerability to the existing statistical watermarks (we do not even need a strong adversary to break the watermarks). We have also included the discussion in the revised paper (See Appendix A.5).

---

> > ### Comment · Reviewer_f25P · 2024-11-26
> >
> > Thank you for the clarification. It would be very valuable to include some version of this response in the main text of the paper (e.g., Related Work section) versus Appendix A.5 since Zhang et al.'s work is closely related. This addresses my questions and I will raise my score accordingly upon seeing the revised version of the paper (in particular, the revised discussion of related work).

---

> > > ### Author Response · Authors · 2024-11-27
> > > **Revised Related Work**
> > >
> > > Thank you for the suggestion. We have revised the related work section to clarify the key differences between our work and that of Zhang et al (See Section 5 of the revised paper). We would also like to further address any additional questions you may have.

---

> ### Comment · Reviewer_f25P · 2024-11-27
>
> I have read the revised paper; the authors have addressed my main concerns re: related work, and I have updated my score accordingly. One lingering point I do not find convincing is the authors' justification for why they focus only on attacking the red-green list watermark of Kirchenbauer et al. versus other distortion-free watermarks that do not change the token distribution. In practice the top-k probabilities may not be available, in which case it is not clear whether the methods generalize to distortion-free watermarks (in particular, are they still more effective than Zhang et al., who do attack these watermarks?).

---

> > ### Author Response · Authors · 2024-12-01
> > **Justification on our focus**
> >
> > Thank you for your response.
> >
> > We believe there is a misunderstanding.
> >
> > First, our assumption easily holds in practical settings--top-k probabilities and the corresponding tokens are available on OpenAI API. Distortion-free watermark schemes, under this setting, are vulnerable to the adversary, who can sample directly from the watermark-free token distribution. Therefore, attacking distortion-free watermarks under our setting is not an interesting/challenging task.
> >
> > Second, although Zhang et al. 's attack does not rely on access to the top-k probabilities, it relies on something that is much more powerful than this information--the attack uses a strong perturbation model that is sometimes larger than the target model (to paraphrase the target text) and an evaluation model to score the paraphrased candidates. Our attack, on the other hand, works well even when it uses a much smaller reference model (e.g., we use TinyLlama-1.3B to attack Llama-7B).
> >
> > We do not claim that our attack will always be better than that of Zhang et al. in all settings. Instead, we claim that, in our threat models, where the adversary has access to the top-k probabilities and the corresponding tokens, we can do better than theirs (although they rely on even stronger models to run their attack). Therefore, the success of our attack conveys a stronger message than theirs. That is, existing statistical watermark schemes are vulnerable to adversaries in practice.

---

### Author Response · Authors · 2024-11-22
**Clarification of threat model**

Reviewers (Reviewer NfPb, Reviewer a4FL) raised concerns about the access of the adversary to the target model.
In our attack, the adversary only has access to the top-K probabilities and the corresponding tokens (with K=20) from the target model at each token position instead of logits/probabilities over all the tokens.

We would like to note that this is a reasonable assumption for the adversary, who could be a user of the target LLM, as common LLM service providers often grant their users access to such information through APIs, e.g., OpenAI’s API, for better customization, interpretability, traceability, and ultimately, better user experience.

---

### Author Response · Authors · 2024-11-22
**On the metric of text quality**

Several reviewers (Reviewer NfPb, Reviewer a4FL, and RzfD) have questioned the use of perplexity as a quality metric.

To explain, perplexity (PPL) is widely used in the text watermarking literature (e.g., [1,2,3,4,5,6]) and serves as a standard metric for evaluating text quality. We followed the literature and included perplexity as a metric for quality evaluation. We are aware of alternative metrics for quality measurement, but finding the universally best quality metric is still an open problem in the NLP literature [7].  In what follows, we discuss several alternative metrics suggested by the reviewers and their inapplicability to our work.

- **P-SP** (suggested by Reviewer a4FL): P-SP metric measures cosine similarity between the embeddings of **two texts** (one of them is regarded as the reference text). It was originally proposed under a different context than watermark detection, e.g., evaluating the quality of a translated/paraphrased sentence. If the given sentence looks/means more similar to the ground truth, then P-SP gives a higher score. In our context, there is no reference text to compare with. Instead, an LLM could be used for answering a question to which there is no single deterministic answer, e.g., explaining why a certain company (e.g., Intel) is good. In such scenarios, answers that look very different may be of high quality at the same time, e.g., both "Intel pays well" and "Intel's CPU plays an important role in commercial computers" are valid answers.

- **BLEU** (suggested by Reviewer NfPb): Similar to P-SP, this metric assumes the existence of a reference text and evaluates its word-level similarity (counting overlaps over n-grams) with a given text. Hence, this metric does not seem to be a proper metric in our setting.

We also run additional experiments using **GPT-4 as the quality metric** to evaluate the generated texts on a scale from 1 to 10 (higher means better). The prompt template is similar to [8] (see Appendix A.4 of the revised version). We present the new results below (also included in Table 1 of the revised version).

| Setting                  | OPT-1.3b (KGW) | OPT-1.3b (Unigram) | Llama2-7b (KGW) | Llama2-7b (Unigram) |
|--------------------------|--------------|------------------|---------------|-------------------|
| Human-written           | 8.83         | 8.83             | 8.83          | 8.83             |
| Reference               | 7.16         | 7.16             | 3.3           | 3.33             |
| Unwatermarked           | 8.66         | 8.66             | 8.66          | 8.66             |
| Watermarked             | 8.33         | 7.66             | 8.28          | 8.33             |
| Watermarked (P-GPT3.5)  | 8.66         | 8.33             | 8.83          | 9.0              |
| Watermarked (Word-D)    | 2.0          | 2.0              | 2.48          | 2.17             |
| Watermarked (Word-S)    | 2.66         | 2.66             | 3.81          | 4.0              |
| Watermarked (Smoothing) | 7.33         | 7.33             | 5.25          | 4.5              |

Our attack achieves comparably good text quality, which is consistently better than the reference model and the Word-D and Word-S attacks that do not use reference models. Notably, on Llama2-7b, the text quality of our attack is much better than the above baselines, and is *only slightly worse* than the original watermarked/unwatermarked models and the unrealistic paraphrasing attack P-GPT-3.5. We note that the evaluation by GPT-4 also has limitations [9, 10, 11,12], e.g., it may be biased toward its own responses [8]. Addressing these limitations is an avenue for future work.

We would like to thank the reviewer for the instructive comments.

---

> ### Author Response · Authors · 2024-11-22
> **On the metric of text quality -- Continued**
>
> [1] Kirchenbauer, John, et al. "A watermark for large language models." ICML, 2023.
>
> [2] Zhao, Xuandong, et al. "Provable Robust Watermarking for AI-Generated Text." ICLR, 2024.
>
> [3] Liu, Aiwei, et al. "An unforgeable publicly verifiable watermark for large language models." *ICLR,* 2023.
>
> [4] Lu, Yijian, et al. "An Entropy-based Text Watermarking Detection Method." ACL, 2024.
>
> [5] Liu, Yepeng, and Yuheng Bu. "Adaptive Text Watermark for Large Language Models." ICML, 2024.
>
> [6] Wu, Yihan, et al. "A Resilient and Accessible Distribution-Preserving Watermark for Large Language Models." ICML, 2024.
>
> [7] Chang, Yupeng, et al. "A survey on evaluation of large language models."  TIST, 2024.
>
> [8] Jovanović, Nikola, Robin Staab, and Martin Vechev. "Watermark Stealing in Large Language Models." ICML, 2024.
>
> [9] Panickssery, Arjun, Samuel R. Bowman, and Shi Feng. "Llm evaluators recognize and favor their own generations." 2024.
>
> [10] Gao, Mingqi, et al. "Llm-based nlg evaluation: Current status and challenges." 2024.
>
> [11] Chu, KuanChao, Yi-Pei Chen, and Hideki Nakayama. "A Better LLM Evaluator for Text Generation: The Impact of Prompt Output Sequencing and Optimization." 2024.
>
> [12] Li, Zhen, et al. "Leveraging large language models for nlg evaluation: Advances and challenges."EMNLP, 2024.

---

### Meta-Review · Area_Chair_Zijr · 2024-12-21

**Metareview:**

**Paper Summary:**

The paper proposes an attack on distortionary text watermarks, using text from an unwatermarked weak LM to rewrite text from a strong watermarked LM in a way that erases the watermark.

**Strengths:**

The attack is interesting and it is simple: a token-level intervention, as opposed to more complicated paraphrasing attacks studied in prior work, e.g., Zhang et al. (ICML,  2024).

**Weaknesses:**

Reviewers a4FL and NfPb note that the attack relies on access to top-k probabilities of the watermarked language model. These are often, but not always available.

Reviewers f25P, NfPb, and RzfD observe that this attacked is not applicable to distortion-free watermarks.

**Additional Comments On Reviewer Discussion:**

Concerns were raised in discussion about the novelty of the proposed method in comparison to previously-proposed paraphrasing attack, e.g., Zhang et al. (ICML, 2024). In each case, a smaller model is used to remove a watermark from a more powerful model. This was largely addressed in discussion.

I have lingering concerns about how Zhang et al. is addressed in the main text. E.g., around line 114:

"rules out the paraphrasing attacks that leverage a strong LM (e.g., ChatGPT) to paraphrase the watermarked text (Zhang et al., 2024a)"

I do not think this is fair to Zhang's work, which really is about by using weaker LMs to attack stronger watermarked LMs. While the revised discussion of Zhang et al. in A.5 is much appreciated and I think the main text could still be improved.

Concerns about applicability to distortion-free watermarks were addressed by the authors definitionally:

> Distortion-free watermark schemes, under this setting, are vulnerable to the adversary, who can sample directly from the watermark-free token distribution. Therefore, attacking distortion-free watermarks under our setting is not an interesting/challenging task.

I am not entirely convinced by this argument. First, to bypass the watermark in this way would require all the model's logits, not just top-k. Second, it doesn't address the point that methods like Zhang et al. really do apply to a broader context.

I do not think that any of these concerns are fatal flaws of this paper. But I do wish these presentational issues had been more satisfactorily resolved during the reviewing period.

---

### Decision · Program_Chairs · 2025-01-22

Reject